# Front-biased activation of the Ras-Rab5-Rac1 loop coordinates collective cell migration

Yuya Jikko[1], Eriko Deguchi[1], Kimiya Matsuda[2], Naoya Hino[3], Shinya Tsukiji[4], Michiyuki Matsuda[1,2,5] and Kenta Terai[1,6,*]

## ABSTRACT

Collective cell migration is coordinated by the front-to-rear intercellular propagation of EGFR-Ras-ERK pathway activation. However, the molecular mechanisms integrating front-to-rear information into this intercellular signaling cascade, particularly the determinants of cellular front-side specification, remain elusive. We visualized the activity of EGFR, Ras, Rac1 and Rab5A (hereafter Rab5) by using FRET biosensors and chemogenetic tools. Whereas EGFR activation was uniformly observed within cells, Ras activation was biased to the front side within cells. The polarized Ras activation depended on Merlin and Rac1, which also showed front-biased activation. Furthermore, Rab5, a crucial regulator of cell migration, demonstrated similar front-biased activation and was found to function downstream of Ras while being necessary for Rac1 activation. Thus, the positive feedback loop consisting of Ras, Rab5 and Rac1 is activated primarily at the front of collectively migrating cells. These findings offer new spatio-temporal insight into processing front–rear information during collective cell migration.

KEY WORDS: Ras protein, Rac1, Rab, Epidermal growth factor receptor, EGFR, ERM family protein, Collective cell migration, Cell migration, Cell polarity

## INTRODUCTION

Wound healing by epithelial cells has been studied as a prototypical example of collective cell migration (Mayor and Etienne-Manneville, 2016). Upon release from confinement, the leader cells facing the open space move forward, guiding the follower cells. Recent studies have shown that parallel waves of mechanical forces and chemical signals arise from the leader cells and propagate to the follower cells (Aoki et al., 2017; Das et al., 2015; Hino et al., 2020; Hiratsuka et al., 2015; Serra-Picamal et al., 2012). This mechano-chemical wave

[1]Department of Pathology and Biology of Diseases, Graduate School of Medicine, Kyoto University, Yoshida-Konoe-Cho, Sakyo-ku, Kyoto 606-8501, Japan. [2]Center for Living Systems Information Sciences, Graduate School of Biostudies, Kyoto University, Yoshida-Konoe-Cho, Kyoto 606-8501, Japan. [3]Institute of Science and Technology Austria, 3400 Klosterneuburg, Austria. [4]Department of Nanopharmaceutical Sciences, Nagoya Institute of Technology, Gokiso-cho, Showa-ku, Nagoya, Aichi 466-0061, Japan. [5]Institute for Integrated Cell-Material Sciences, Kyoto University, Yoshida-Ushinomiya-cho, Sakyo-ku, Kyoto 606-8317, Japan. [6]Department of Anatomy and Cell Biology, Graduate School of Biomedical Sciences, Tokushima University, Kuramoto-cho, Tokushima 770-8503, Japan.

*Author for correspondence (terai.kenta.5m@tokushima-u.ac.jp)

S.T., 0000-0002-1402-5773; M.M., 0000-0002-5876-9969; K.T., 0000-0001-7638-3720

functions to instruct the follower cells in the front–rear direction and facilitates collective movement (Hirashima et al., 2023; Ladoux et al., 2016). Of note, the ligands of the epidermal growth factor receptor (EGFR) are a principal chemical mediator during the collective cell migration (Lin et al., 2021). The EGFR ligands activate EGFR and thereby trigger the Ras-ERK (ERK1 and ERK2, also known as MAPK3 and MAPK1, respectively) signaling cascade, culminating in the activation of the ADAM17 matrix metalloproteinase. ADAM17 in turn cleaves off pro-EGFR ligands on the plasma membrane, triggering the next cycle of EGFR activation (Aoki et al., 2017, 2013).

Small GTPases play pivotal roles in the establishment of front–rear polarization of migrating cells. During chemotaxis of *Dictyostelium discoideum* cells in response to cAMP, Ras is activated at the front of the migrating cells, where phosphoinositide-3-kinase (PI3K)-dependent actin polymerization drives lamellipodia and pseudopodia protrusion (Nakajima et al., 2014; Sasaki et al., 2004; Xiong et al., 2010). Meanwhile, in mammalian migrating cells, lamellipodia and filopodia are induced at the front by Rac1 and Cdc42, respectively (Hall, 1998; Mitchison and Cramer, 1996). In support of this observation, activation of Rac1 and Cdc42 in the lamellipodia of migrating cells has been visualized by biosensors based on the principle of Förster resonance energy transfer (FRET) (Itoh et al., 2002; Kraynov et al., 2000; Kurokawa et al., 2004). Another small GTPase that is involved in the cell migration is Rab5A, a regulator of early endosome (Sigismund et al., 2021). Overexpression of Rab5A induces coordinated motility of cells by increasing non-clathrin-dependent internalization of EGFR (Malinverno et al., 2017; Palamidessi et al., 2019). However, the front–rear bias of Rab5A activity has not been known.

The aim of this study is to elucidate the molecular mechanisms that coordinate the front-to-rear intercellular propagation of EGFR-Ras-ERK pathway activation during collective cell migration, particularly identifying the determinants of cellular front-side specification. To address this, we visualized the activity of EGFR, Ras, Rac1 and Rab5A (hereafter Rab5) using FRET biosensors in Madin-Darby Canine Kidney (MDCK) cells. The front–rear polarity of activity was detected in Ras, Rac1 and Rab5, but not in EGFR, excluding the gradient of EGFR ligand as the generator of cell polarity. Meanwhile, pulsatile activation waves from the leader cells were observed for both EGFR and Ras. These observations suggest that EGFR ligands, which are chemical signals, only convey information about timing, whereas the information about the front and back of the cell is controlled by Rac1. Furthermore, Rab5 activity exhibited a front–rear gradient in migrating cells, which is essential for the Rac1 activation in the cell front. In summary, Ras and Rab5 serve as the integrators of these temporal and spatial inputs.

## RESULTS

### Ras, but not EGFR, is activated at the front of the follower cells

During collective cell migration, the ERK waves direct front–rear polarity within cells (Hino et al., 2020). However, it has not been

examined whether the activity of the EGFR-Ras-ERK signaling cascade is also polarized. To answer this challenging question, we examined the subcellular activities of EGFR and Ras by using Picchu-X and Raichu-Ras FRET biosensors, respectively. Picchu-X is composed of the CRK adaptor protein sandwiched by the cyan fluorescent protein (CFP) and yellow fluorescent protein (YFP) (Kurokawa et al., 2001) (Fig. S1A). Because Crk is phosphorylated by EGFR, the plasma membrane-targeted Picchu-X can be used as a surrogate marker of the EGFR activity at the plasma membrane. Raichu-Ras, which is made up of Ras and the Ras-binding domain of Raf, is used to monitor the GTP-to-GDP ratio on Ras (Mochizuki et al., 2001) (Fig. S1B). We also used cells expressing the ERK biosensor EKARrEV-NLS as reported previously (Lin et al., 2021) (Fig. S1C). These biosensor-expressing cells were seeded in a culture insert placed on a collagen-coated cover glass. After an overnight incubation, the confinement was released to drive collective cell migration. We observed propagation of EGFR activation and Ras activation as well as ERK activation (Fig. 1A–C; Movie 1). Kymograph analysis revealed that the EGFR and Ras were activated almost synchronously with ERK (Fig. 1D–F).

We encountered two problems in the conventional confinement release assay. First, the FRET biosensors for EGFR and Ras were expressed evenly at the plasma membrane and looked to be enriched at the cell-to-cell border where the follower cells extended cryptic lamellipodia below the leading cells (Fig. 1G). Under the confluent conditions, however, we were unable to segregate signals of the two overlapping plasma membranes. Second, the methods used to perturb intracellular signaling molecules often inhibit collective cell migration. Therefore, it is often difficult to know whether the observed effects are attributable to the intracellular events or simply to the absence of external cues to the cells. To overcome these problems, we plated the FRET biosensor-expressing MDCK cells and ERK-KTR-expressing MDCK cells (see below for explanation of the construct) at a 1:10 ratio in a silicone chamber and triggered collective cell migration upon confinement release (Fig. 1H). In this way, the signals at the front of the FRET biosensor-expressing cells could be segregated from those at the rear of the cells, and even cells with migration defects could be brought forward along with the surrounding wild-type cells. ERK-KTR–mCherry (Regot et al., 2014) moves from the nucleus to the cytoplasm upon phosphorylation by ERK, for the simultaneous observation of ERK activity with the CFP/YFP-based FRET biosensors (Fig. S2A,B). To quantify the front–rear polarity of the EGFR and Ras activities, we defined the polarity index as described in the Materials and Methods section (Fig. 1I). Briefly, we analyzed clusters of FRET biosensor-expressing cells 10 h after the release of confinement. The direction of cell movement, which is usually close to the direction of group migration or the $x$-axis, was determined by using the Trackmate program. Using a line perpendicular to the direction of cell movement, the cell area was divided into front and rear zones. Meanwhile, cell clusters were delineated to extract the plasma membrane regions that were in contact with the neighboring cells without FRET biosensor expression. With these masks, we calculated the FRET/CFP fluorescent ratio at the plasma membrane of the front and rear zones. The polarity index was defined by the activity ratio at the front versus that at the rear plasma membranes. Furthermore, the clusters of FRET biosensor-expressing cells were categorized into ERK wave (+) and ERK wave (−) clusters according to the ERK activity of cells surrounding the clusters (Fig. S2C,D).

Because EGFR ligands mediate the ERK wave propagation, we anticipated the concentration gradient of EGFR ligands, which would cause front–rear polarization of EGFR activity. For example,

in *Drosophila* border cells, which also migrate collectively in an EGFR-dependent manner, EGFR was activated primarily at the front of migrating cells (Assaker et al., 2010; Jékely et al., 2005). However, against our expectation, we were not able to detect front–rear bias of EGFR activity in either the ERK wave (+) or ERK wave (−) clusters (Fig. 1J,K; Movie 2). During collective cell migration, ERK waves propagate at a velocity of 2 μm/min with a full width at half maximum (FWHM) of 30 min (Aoki et al., 2017). Assuming that the size of a cell is 30 μm and that the wave velocity is determined by the diffusion of EGFR ligands, we should have been able to observe an intracellular gradient of EGFR activity. This observation suggests that, in addition to the binding to EGFR ligands, alterations in mechanical force are required for EGFR activation (Hino et al., 2020), and such changes in mechanical force occur both at the front and at the rear plasma membrane without significant time delay. In stark contrast, Ras was activated primarily at the front of migrating cells, presumably at the cryptic lamellipodia (Fig. 1L,M; Movie 2). The polarity was observed within the same clusters, both in the ERK wave (+) and ERK wave (−) conditions, but was enhanced in the ERK wave (+) clusters (Fig. 1M, right panel). Thus, we concluded that the front–rear polarization occurs at the level of Ras, but not EGFR.

### SOS is the primary factor responsible for front-biased Ras activation

We next aimed to elucidate the mechanism underlying the front-biased Ras activation. To exclude the possibility that the front-biased activation was generated by the gradient of EGFR ligand concentration, EGFR was activated by the bath application of EGF into the medium during collective cell migration (Fig. 2A,B). Cells were serum-starved from the onset of migration to exclude the effect of growth factors in the medium. Cells kept moving with ERK waves even under the serum-starved condition. Cells in the ERK wave (−) clusters did not show a significant front–rear bias of Ras activity but exhibited front-biased Ras activation upon EGF addition. In the ERK wave (+) clusters, the front-biased Ras activation was observed irrespective of the exogenous EGF. Thus, Ras activation occurs preferentially at the front.

Activated EGFR recruits the adaptor protein Grb2 in complex with the SOS guanine nucleotide exchange factor to the plasma membrane. Subsequently, the autoinhibitory domain of SOS is unlocked through the interaction with phosphatidylinositol 4,5-bisphosphate (PIP$_2$; Chen, 1997; Gureasko et al., 2008), ezrin, radixin or moesin (ERM) proteins (Geißler et al., 2013) or Ras (Margarit et al., 2003), culminating in the Ras activation. Thus, we hypothesized that the SOS1-biased translocation causes the front-biased activation. To test this model, the self-localizing ligand-induced protein translocation (SLIPT) assay (Suzuki et al., 2022) was used to eliminate the biased translocation. In this system, the myristoylated ligand, m$^D$cTMP, induces the translocation of internally K6-tagged dihydrofolate reductase ($^{iK6}$DHFR)-fused proteins to the plasma membrane. When we expressed a chemogenetic tool to relocate Grb2 (chGrb2/$^{iK6}$DHFR-iRFP713), in which the SH2 domain of Grb2 was replaced with $^{iK6}$DHFR (Fig. 2C), m$^D$cTMP addition recruited chGrb2/$^{iK6}$DHFR-iRFP713 to the plasma membrane and elevated Ras activity (Fig. S3A). Then, we applied m$^D$cTMP to the collectively migrating cells after serum starvation (Fig. 2D,E). The effect of m$^D$cTMP was a mirror image of the effect of EGF. Namely, Ras was activated irrespective of the phase of the ERK waves (Fig. 2E, left panel), and the front-biased Ras activation was induced in the ERK wave (−) clusters (Fig. 2E, right panel). This observation suggests that the front-biased Ras activation might occur at the level of SOS. To further

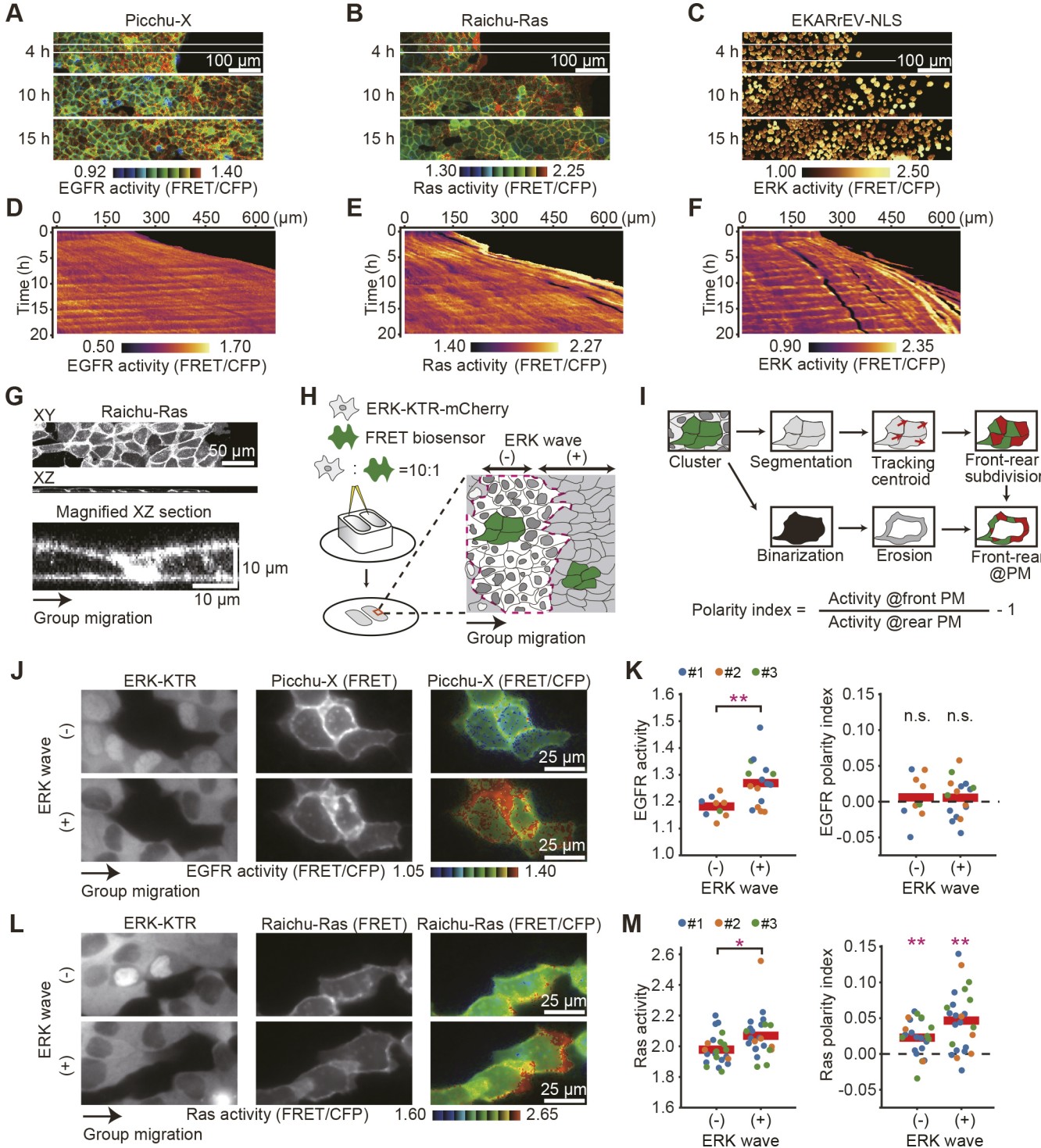

**Fig. 1. Front-biased activation of Ras, but not EGFR, in collectively migrating cells.** (A–C) MDCK cells expressing Picchu-X (A), Raichu-Ras (B) or EKARrEV (C) were subjected to a confinement release assay under 10% FBS conditions. FRET/CFP ratio images show the EGFR, Ras and ERK activity. Representative images at 4, 10 and 15 h after the release are shown. Scale bar: 100 μm. (D–F) X-T kymographs of the FRET/CFP ratio images in A–C. Kymographs were taken from the areas indicated by the white boxes in panels A to C. (G) *xy* (upper) and *xz* (middle) sections of the *z*-stack fluorescent images of Raichu-Ras. A magnified *xz* image (beneath) is also shown. Scale bars: 50 μm and 10 μm, respectively. The images in A–G are representative of a single experiment. (H,I) Schematics of the confinement release assay and the quantification. FRET biosensor-expressing MDCK cells and ERK-KTR–mCherry-expressing MDCK cells were mixed and seeded at a 1:10 ratio into a silicone chamber. Collective cell migration was started upon confinement release. (J) ERK-KTR–mCherry image, FRET channel image and FRET/CFP ratio image of MDCK cells expressing ERK-KTR–mCherry or Picchu-X. Scale bar: 25 μm. (K) The EGFR activity represented by the FRET/CFP ratio (left) and the EGFR polarity index (right) in the ERK wave (−) and (+) clusters. The three colors correspond to independent experiments. The red bars indicate the means. A total of 26 cell clusters were studied. \*\**P*<0.001; n.s., not significant (*P*>0.2) (unpaired two-tailed Welch's *t*-test). The polarity index values were subjected to statistical comparison against zero with the same sample size, focusing on polarity detection within individual samples rather than comparing differences between samples. (L,M) Ras activity (left) and polarity index (right) are represented as in J and K. *n*=47 cell clusters. \**P*<0.01; \*\**P*<0.001 (unpaired two-tailed Welch's *t*-test).

Journal of Cell Science

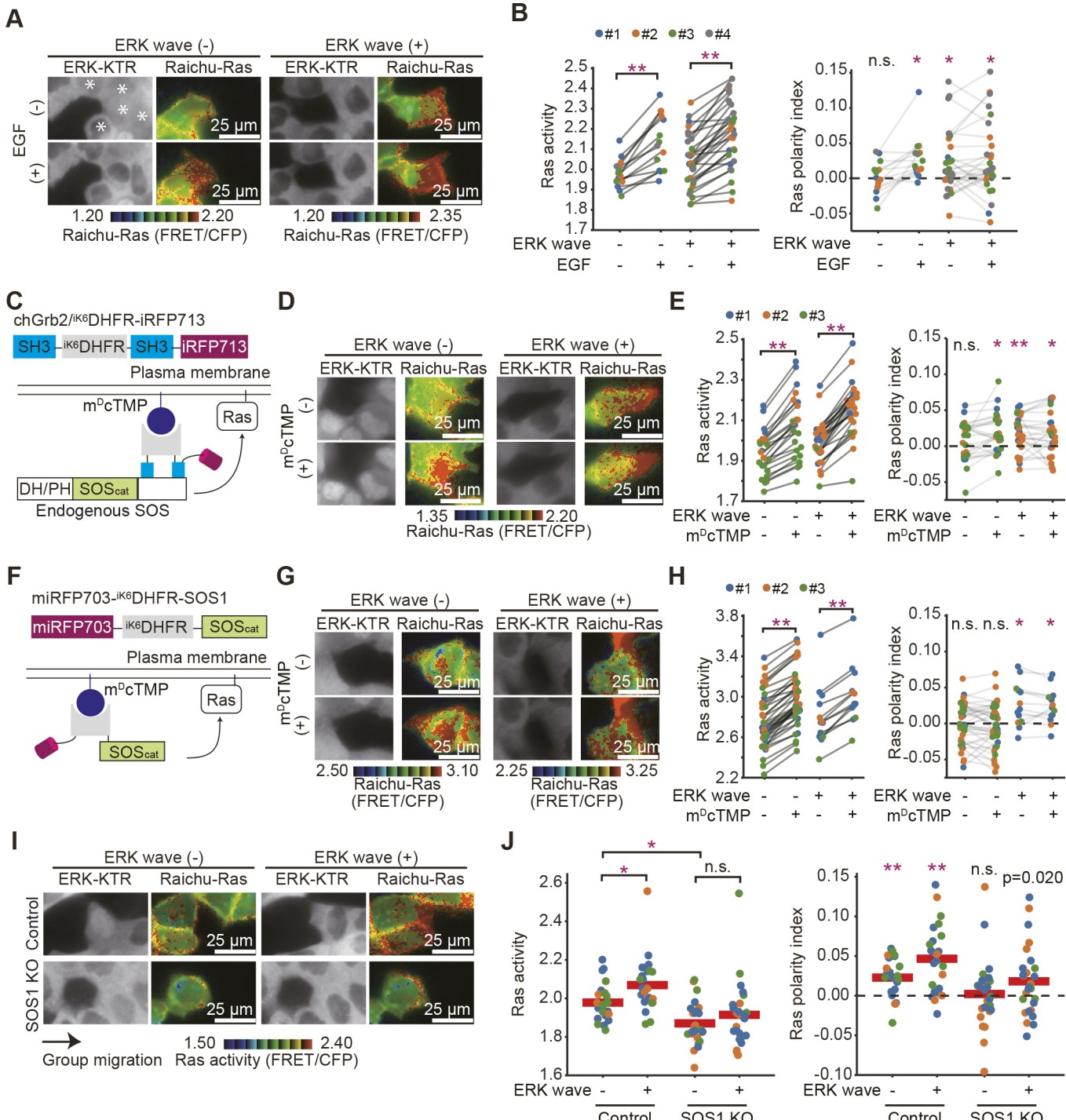

**Fig. 2. Front-biased Ras activation induced by plasma membrane translocation of Grb2.** (A) MDCK cells expressing Raichu-Ras were mixed with those expressing ERK-KTR–mCherry in a 1:10 ratio and subjected to the confinement release assay without FBS. Shown are images 5 min before and 5 min after the application of 2 ng ml⁻¹ EGF. White asterisks indicate the non-phosphorylated ERK-KTR in the nuclei, which are the hallmarks of the ERK wave (+) clusters. (B) Quantified data for the clusters in A. Each dot corresponds to a single cluster. Gray lines represent the same cluster. Data from four independent experiments are combined. A total of 43 cell clusters were studied. *$P$<0.01; **$P$<0.001; n.s., not significant ($P$>0.05) (paired $t$-test for Ras activity; unpaired two-tailed Welch's $t$-test for polarity index). The values of the polarity index were subjected to statistical comparison against zero with the same sample size. (C) A schematic of the SLIPT assay using chGrb2/iK6DHFR-iRFP713. chGrb2/iK6DHFR-iRFP713 is made of the SH3 domains of Grb2, iK6DHFR and iRFP713. Upon the mDcTMP treatment, chGrb2/iK6DHFR-iRFP complexed with endogenous SOS is recruited to the plasma membrane. (D) MDCK cells expressing chGrb2/iK6DHFR-iRFP713 and Raichu-Ras were mixed with those expressing ERK-KTR–mCherry at a 1:10 ratio and subjected to the confinement release assay as in A. Representative images before and after the addition of 2 µM mDcTMP are shown. (E) Quantified data for the clusters in D. $n$=51 cell clusters. Statistical significance was determined as in B. (F) A schematic of miRFP703-eDFHR-SOS_cat, which comprises miRFP703, iK6DHFR and the catalytic domain of SOS. (G) Confinement release assay performed as in A. Representative images before and after the addition of 20 nM mDcTMP are shown. (H) Quantified data for the clusters in G. A total of 52 cell clusters were studied. Statistical significance was determined as in B. (I) MDCK cells and the SOS1-knockout MDCK cells expressing Raichu-Ras were used for the confinement release assay as in Fig. 1L. (J) Quantified data for the clusters in (I). Red lines represent the mean values. Note that the control data are from Fig. 1M. A total of 42 (SOS1 KO) cell clusters were studied. Statistical significance was determined as in B.

examine this possibility, we used miRFP703-$^{iK6}$DHFR-SOS$_{cat}$, in which only the catalytic domain of SOS was fused to $^{iK6}$DHFR (Fig. 2F). Expression of the catalytic domain of SOS increased Ras activity, which was further enhanced by m$^{D}$cTMP stimulation (Fig. S3B; Fig. 2G,H). In stark contrast to cells expressing chGrb2/$^{iK6}$DHFR-iRFP713, the front-biased activation did not appear after m$^{D}$cTMP addition in the ERK wave (−) clusters (Fig. 2H, right panel). These data suggest that the SOS autoinhibition is released preferentially at the front of cells.

We next examined the phenotypes of SOS-knockout MDCK cells (Fig. 2I,J). In SOS1-knockout cells (Fig. S4A), Ras activity was reduced, and its front-biased activation was partially attenuated. It is possible that this partial effect is attributed to the presence of SOS2, the other SOS protein that remains intact in SOS1-knockout MDCK cells. We were unable to establish SOS1/2 double-knockout cell lines, likely due to their essential role in cell growth.

### Rac1 and Merlin signaling is required for the front-biased Ras activation

We then hypothesized that the lamellipodia is the crucial structure for releasing the SOS autoinhibition and activating Ras at the leading edge. To test this hypothesis, we investigated the role of Rac1, a key regulator of lamellipodia formation in front-biased Ras activation. The expression of a dominant-negative Rac1 mutant or Rac1 knockdown (Fig. S4B) suppressed Ras activity, resulting in the disappearance of front-biased activation in both ERK wave (+) and ERK wave (−) clusters (Fig. 3A,B).

We next examined the role of Merlin, given its function in regulating Rac1 during collective cell migration as well as suppressing EGFR and antagonizing the ERM proteins (Cole et al., 2008; Das et al., 2015; Geißler et al., 2013). In Merlin-deficient cells, Ras activity did not change significantly; however, the front-biased Ras activation disappeared (Fig. 3C,D). This effect was largely rescued by the re-expression of Merlin (Fig. S4C, third column from the right in Fig. 3D). Thus, the front-biased Ras activation depends on Rac1 and Merlin. We also attempted to visualize the translocation of Merlin tagged with EGFP during collective cell migration. However, the attempt was unsuccessful, possibly because the EGFP tag interfered with the proper localization or dynamics of Merlin during collective migration. Additionally, the spatiotemporal regulation of Merlin might be too subtle or transient to be captured under our imaging conditions.

### Front-biased Rac1 activity depends on Rab5

Next, Rac1 activation during collective cell migration was visualized with the Raichu-Rac1 biosensor (Fig. 4A,B, Movie 3). As reported previously (Kurokawa and Matsuda, 2005) (Fig. S1D), Rac1 showed high activity at the lamellipodia in the leader cells, but the wave propagation was not as discrete as detected for ERK and Ras in the follower cells. In the presence of non-fluorescent cells, however, Rac1 was activated as the front-biased activity in the ERK wave (+) clusters (Fig. 4C,D, Movie 4). Thus, the ERK wave does not significantly increase the net Rac1 activity but instructs cells in the front–rear bias. In Merlin knockout cells, the front–rear bias in the ERK wave (+) clusters did not appear to be affected (Fig. 4D, right), while the front–rear axis markedly deviated from the direction of group migration (Fig. 4E), suggesting that Merlin is required to orient the direction of Rac1-dependent membrane protrusion.

We next examined the contribution of Rab5, which is known to regulate Rac1 and thereby cell migration of MDCK cells (Sigismund et al., 2021). Expression of Rab5S34N, a dominant-negative mutant, markedly decreased Rac1 activity and abolished

the front–rear polarity (Fig. 4C,D), suggesting that Rac1 functions downstream of Rab5 during collective cell migration.

### The positive feedback loop composed of Ras, Rab5 and Rac1 for the front–rear polarization

We extended our visualization to Rab5 by using the Rab5 biosensor, Raichu-Rab5 (Fig. S1E), which is anchored to the plasma membrane to measure activity just beneath the membrane and exclude vesicular signals (Fig. S5). The waves from the leader cells were not clear (Fig. 5A,B; Movie 5). In the presence of non-fluorescent cells, like Rac1 activity, high Rab5 activity was restricted to primarily the lamellipodia in the ERK wave (+) clusters (Fig. 5C,D, Movie 6). The front-biased Rab5 activation during collective cell migration was abolished by the expression of dominant-negative Ras, suggesting that Rab5 operates downstream of Ras. In support of this view, recent studies have shown that the overexpression of Rab5 causes unjamming of the epithelial monolayer and large-scale coordination of anisotropic cell motility (Malinverno et al., 2017; Palamidessi et al., 2019).

How does the dominant-negative Rac1 inhibit the front-biased activation of Ras? Since SOS is crucial for driving front-biased Ras activity, we hypothesize that localized activation of Rac1 facilitates the translocation of SOS to the plasma membrane. We attempted to observe the localization changes of SOS using a GFP tag; however, the results were inconclusive. Alternatively, Rac1 might accelerate the guanine nucleotide exchange activity of SOS. Further optimization of the experimental conditions is therefore necessary to accurately assess both the translocation dynamics and activity change of Sos.

### Collective cell migration is coordinated by SOS, Ras, Rac1 and Rab5

What is the impact of each molecule on collective cell migration? We analyzed cells lacking SOS1 or expressing dominant-negative mutants of Ras, Rab5 or Rac1, as shown in Fig. 6. In this assay system, cells located 0–0.3 mm from the edge were regarded as leader cells, whereas those at 0.6–1.2 mm were classified as follower cells. The migration speed of cells expressing RasS17N and Rac1T17N was attenuated in both leader and follower cells, whereas follower cell migration was reduced under all mutant conditions. Notably, leader and follower cells exhibit distinct characteristics – leaders migrate toward free space, whereas followers migrate via ERK wave propagation, as demonstrated by Hino et al. (Hino et al., 2020). Given that Ras and Rac activation has been shown to be essential for collective cell migration (Aoki et al., 2017), we focused on the unique phenotypes of cells expressing Rab5S34N and those lacking SOS1. In Rab5S34N cells, ERK wave propagation was absent (Fig. 6A), suggesting that migration directionality was disrupted. This is consistent with the role of ERK waves in determining the direction of cell migration (Hino et al., 2020). Accordingly, the directedness (see Fig. 6C) of migration in Rab5S34N-expressing follower cells was randomized (Fig. 6D). How about the impact of SOS1 removal? SOS1 depletion led to a reduced migration speed while maintaining intact directedness. However, the rhythm of the ERK wave was elongated (Fig. 6E) and the propagation of ERK activation was delayed (Fig. 6F,G), which likely contributed to the reduced migration speed.

### DISCUSSION

This study has shown the positive feedback loop activation of the three small GTPases, Ras, Rab5 and Rac1, that coordinate collective cell migration (Fig. 6E). We have demonstrated that the front-side

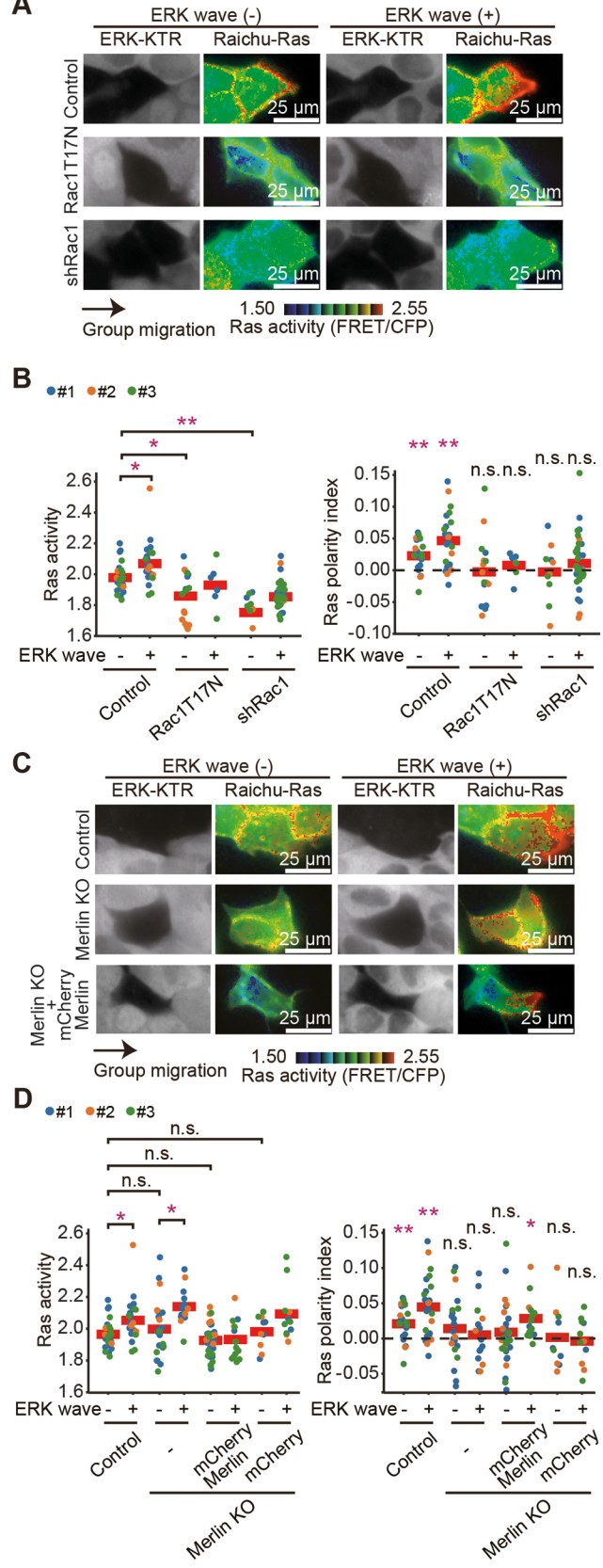

**Fig. 3. Rac1 and Merlin signaling is required for the front-biased Ras activation.** (A) MDCK cells expressing Raichu-Ras with or without Rac1T17N or shRac1 were mixed with those expressing ERK-KTR–mCherry in a 1:10 ratio and subjected to the confinement release assay as described in Fig. 1L. (B) Red lines represent the mean values. Note that the control data are from Fig. 1M. A total of 23 (Rac1T17N) and 44 (shRac1) cell clusters were studied. Statistical significance was determined as in Fig. 1M. *P<0.01; **P<0.001; n.s., not significant (P>0.05). (C) MDCK cells or Merlin-knockout MDCK cells with or without mCherry–Merlin expressing Raichu-Ras were subjected to the confinement release assay. (D) Quantified data for the clusters in (C). Red lines represent the mean values. Note that the control data are from Fig. 1M. A total of 37 (Merlin KO), 50 (Merlin KO with mCherry-Merlin), and 21 (Merlin KO with only mCherry) cell clusters were studied. Statistical significance was determined as in Fig. 1M.

present, as shown in Fig. S1A,B. Although ERK wave propagation was unclear in the kymographs of Raichu-Rac1- and Raichu-Rab5-expressing cells, the biased activity, as indicated by the polarity index in Figs 4D and 5D, was exaggerated only in the ERK wave-positive clusters, demonstrating the synchrony of the activation rhythm between ERK and Rac1, as well as ERK and Rab5.

Three mechanisms, which are not mutually exclusive, might underlie the front-biased SOS-dependent Ras activation. First, lamellipodial accumulation of $PIP_2$, which releases the autoinhibition of SOS (Chen, 1997; Gureasko et al., 2008), might cause Ras activation at the front. In fact, the $PIP_2$ concentration is higher in the front than in the rear of migrating MDCK cells (Nishioka et al., 2008). Second, lamellipodia at the front might be enriched by ERM proteins, which enhance SOS catalytic activity (Geißler et al., 2013). The inhibitory effect of Merlin deficiency on the polarized Ras activation (Fig. 3D) agrees with a report in which ezrin was antagonized by Merlin (Geißler et al., 2013). Third, the lipid raft protein flottilin-1, which promotes the association of SOS with H-Ras and K-Ras, might facilitate front-biased Ras activation (Jin et al., 2023), because lamellipodial protrusions are enriched with lipid rafts (Mañes et al., 1999).

Furthermore, Merlin instructs cells in the direction of group migration. The appearance of clear ERK waves usually takes 8 h. Probably this time is required for the establishment of the front–rear polarity that facilitates collective cell migration via a mechano-chemical wave. Additional studies will be needed to elucidate how and when these polarities are formed after the onset of migration.

Why does SOS1 depletion prolong the ERK wave cycle? We hypothesize that the extended ERK wave cycle observed in SOS1-deficient cells is caused by an elevated threshold for ligand-induced signaling. In the absence of SOS1, greater input is required to activate the RAS–ERK pathway, necessitating increased growth factor accumulation. This delay in reaching the activation threshold postpones downstream signaling and lengthens each ERK cycle. Consistent with this model, the ERK wave propagation speed was markedly reduced in SOS-knockout cells, as demonstrated by kymograph analyses (Fig. 6F,G), which might contribute to the prolonged cycle by reflecting cumulative delays in signal initiation and transmission. These findings implicate SOS1 as a key regulator of the temporal precision of ERK signaling, although further quantitative studies are warranted to validate this mechanism.

## MATERIALS AND METHODS
### Plasmids
The cDNAs encoding Rac1T17N, Rab5S34N, HRasS17N, mCherry–Merlin, mCherry and ERK-KTR–mCherry were subcloned into the pPB vector (Yusa et al., 2009) using Ligation high Ver. 2 (TOYOBO, Osaka, Japan) or the In-Fusion cloning kit (Clontech, Mountain View, CA). The cDNA of mCherry was a gift from Dr Ichiro Nakagawa (Kyoto University, Japan)

biased activation of Ras, Rab5 and Rac1, but not EGFR, occurs during collective migration in MDCK cells, primarily caused by SOS1-biased activation. Supporting this, a synchronized activation rhythm between ERK and Ras, as well as ERK and EGFR, is

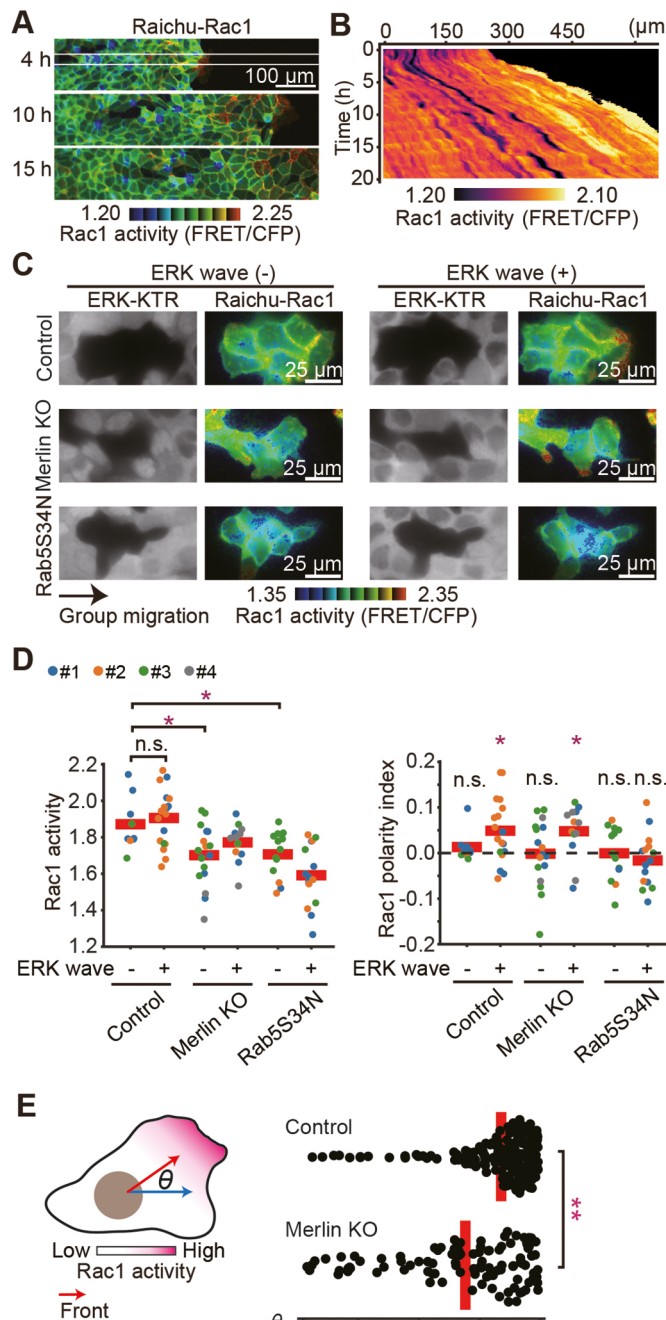

**Fig. 4. Front-biased Rac1 activity depends on Rab5.** (A) MDCK cells expressing Raichu-Rac1 were subjected to confinement release assay as in Fig. 1A. FRET/CFP ratio images show the Rac1 activity. Representative images at 4, 10 and 15 h after release are shown. Scale bar: 100 µm. (B) X-T kymographs of the FRET/CFP ratio images in A. Kymograph was taken from the area indicated by the white box in A. (C) The parent MDCK cells, the Merlin-knockout MDCK cells and the Rab5S34N-expressing MDCK cells that expressed Raichu-Rac1 were mixed with ERK-KTR–mCherry-expressing MDCK cells in a 1:10 ratio and subjected to the confinement release assay as in Fig. 1L. (D) Quantified data for the clusters in C. Each dot corresponds to a single cluster. Data from three independent experiments are shown. Red lines represent the mean values. Results are shown for a total of $n$=27 (Control), $n$=31 (Merlin KO) and $n$=28 (Rab5S34N) cell clusters. *$P$<0.01; **$P$<0.001; n.s., not significant ($P$>0.05) (unpaired two-tailed Welch's $t$-test for Rac1 activity and the polarity index). The values of the polarity index were subjected to statistical comparison against zero with the same sample size. (E) The angles between the group migration and the front of cells are plotted. Note that the control data were analyzed using the same dataset as utilized in Figs 1M and D, and the data of Merlin KO were from the same dataset as in Fig. 3D and D. Results are shown for a total of $n$=202 (control) and $n$=106 (Merlin KO) cells. *$P$<0.01; **$P$<0.001; n.s., not significant ($P$>0.15) (unpaired two-tailed Welch's $t$-test for the angle).

from the original version. The plasmids used in this paper are listed in Table S1A.

## Reagents and antibodies

The following reagents were used: EGF (no. E9644; Sigma-Aldrich, St Louis, MO, USA) and bovine serum albumin (no. A2153; Sigma-Aldrich). m$^D$cTMP was synthesized as described previously (Suzuki et al., 2022).

The following primary and secondary antibodies were used for immunoblotting: anti-β-actin rabbit antibody (no. 4970, Cell Signaling Technology, Beverly, MA, USA; 1:1000 dilution); anti-SOS1 rabbit antibody, CT (no. 07-337, Upstate Biotechnology, Lake Placid, NY, USA; 1:1000 dilution); anti-Merlin rabbit antibody (no. BS3663P, Bioworld Technology, St. Louis Park, MN, USA; 1:1000 dilution); anti-Rac1 mouse antibody (no. 610650, BD Biosciences, Franklin Lakes, NJ, USA; 1:1000 dilution); anti-α-tubulin mouse antibody (no. CP06, La Jolla, CA; 1:1000 dilution); IRDye 680-conjugated goat anti-mouse-IgG antibody (no. 926-32220, LI-COR Biosciences, Lincoln, NE, 1:10,000 dilution); and IRDye 800CW goat anti-rabbit IgG antibody (no. 926-32211, LI-COR Biosciences, 1:10,000 dilution).

## Cell culture

MDCK (ECACC 84121903) cells were obtained from the European Collection of Authenticated Cell Cultures (ECACC) via the RIKEN BioResource Center (no. RCB0995). Lenti-X 293T cells were purchased from Clontech (no. 632180; Mountain View, CA). These cells were maintained in DMEM (no. 044-29765; FUJIFILM Wako Pure Chemical Corporation, Osaka, Japan) supplemented with 10% FBS (F7524; Sigma-Aldrich), 100 units ml$^{-1}$ penicillin and 100 µg ml$^{-1}$ streptomycin (no. 26253-84; Nacalai Tesque, Kyoto, Japan) in a 5% CO$_2$ humidified incubator at 37°C.

## Establishment of stable cell lines

A lentiviral expression system was employed to establish MDCK cells stably expressing EKARrEV-NLS, Raichu-Rab5 with a K-Ras membrane-targeting sequence (KRasCAAX: 5′-AAGATGAGCAAAGATGGTAA-AAAGAAGAAAAAGAAGTCAAAGACAAAGTGTGTAATTATG-3′), or H2B–iRFP as described previously (Hino et al., 2020; Lin et al., 2021). Briefly, for the preparation of the lentivirus, pCSII-EKARrEV-NLS (Lin et al., 2021), pCSIIbsr-Raichu-Rab5/PM or pCSIIbleo-H2B-iRFP, a vector for lentiviral transduction (Miyoshi et al., 1998), was co-transfected with psPAX2 (Addgene no. 12260) and pCMV-VSV-G-RSV-Rev (a gift from Hiroyuki Miyoshi, RIKEN BioResource Center, Japan) into Lenti-X 293T cells by using polyethylenimine (no. 24765-1; Polyscience Inc., Warrington, PA, USA). MDCK cells were incubated with the lentivirus, and after 2 days of incubation the cells were treated with 10 mg ml$^{-1}$ blasticidin S (no. 029-18701; Wako) for bsr or 100 mg ml$^{-1}$ zeocin

(Shaner et al., 2004). To generate pCSIIbsr-Raichu-Rab5/PM, cDNAs coding Raichu-Rab5/PM were subcloned into the pCSII vector using the SLiCE cloning method (Zhang et al., 2014). pCAGGS-chGrb2/$^{iK6}$DHFR-iRFP713 (Addgene no. 178854) was reported previously. cDNA of pCAGGS-chGrb2/$^{iK6}$DHFR-iRFP713 was subcloned into the pPB vector using the In-Fusion cloning kit (Clontech). pPBpuro-miRFP703-$^{iK6}$DHFR-cRaf (Addgene no. 209919) was reported previously. cDNA of miRFP703 was a gift from Kazuhiro Aoki (National Institutes of Natural Sciences, Okazaki, Japan). To generate an expression plasmid encoding miRFP703-$^{iK6}$DHFR-mSOS1-linkercat, cRaf in pPBpuro-EGFP-$^{iK6}$DHFR-cRaf was substituted by mSOS1-linkercat using the In-Fusion cloning kit (Clontech). mSOS1-linkercat was reported previously (Komatsu et al., 2018). Expression plasmids encoding EKARrEV-NLS (Lin et al., 2021), Raichu-Ras (Mochizuki et al., 2001), Raichu-Rab5/PM (Kitano et al., 2008) and RaichuEV-Rac1 (Komatsu et al., 2011) were reported previously. An expression plasmid encoding the FRET tyrosine kinase biosensor, Picchu, was subjected to W169L mutation

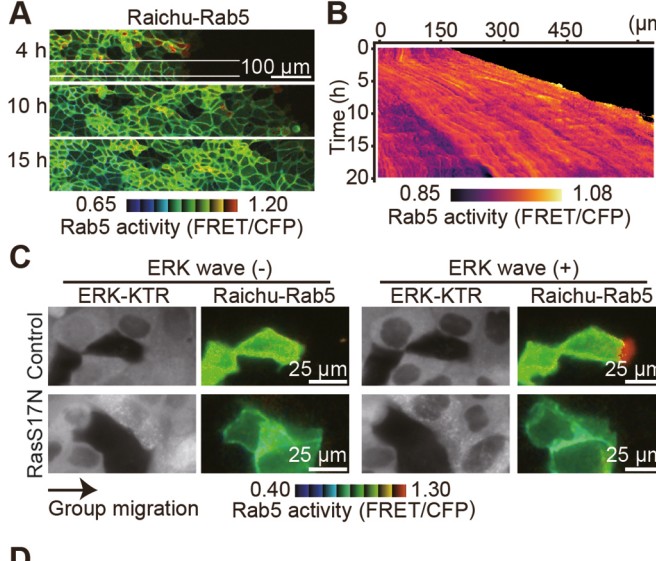

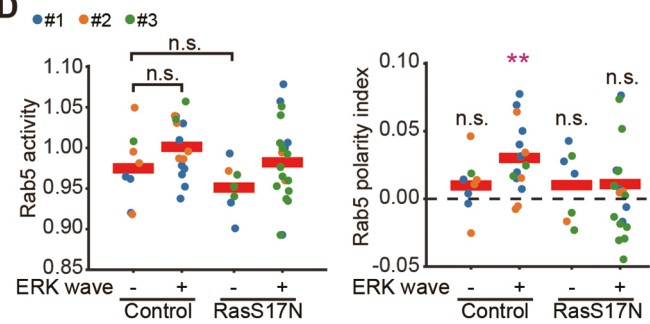

**Fig. 5. The positive feedback loop comprising Ras, Rab5 and Rac1.**
(A) MDCK cells expressing Raichu-Rab5 were subjected to a confinement release assay as in Fig. 1A. FRET/CFP ratio images show the Rab5 activity. Representative images at 4, 10 and 15 h after release are shown. Scale bar: 100 μm. (B) X-T kymographs of the FRET/CFP ratio images in A. Kymograph was taken from the area indicated by the white box in A. (C) The parent MDCK cells and HRasS17N-expressing MDCK cells that express Raichu-Rab5 were mixed with ERK-KTR–mCherry-expressing MDCK cells in a 1:10 ratio and subjected to the confinement release assay as in Fig. 1L. (D) Quantified data for the clusters in C. Each dot corresponds to a cell cluster. Data from three independent experiments are shown. Red lines represent the mean values. Results are shown for a total of $n$=24 (control) and $n$=27 (RasS17N) cell clusters. **$P$<0.001; n.s., not significant ($P$>0.05) (unpaired two-tailed Welch's $t$-test for Rab5 activity and the polarity index). The values of the polarity index were subjected to statistical comparison against zero with the same sample size.

(no. 11006-33-0; InvivoGen, San Diego, CA, USA) for bleo for the selection. The MDCK cells expressing Raichu-Rab5/PM were sorted using a FACS Aria IIu cell sorter (Becton Dickinson, Franklin Lakes, NJ) with CFP and FRET fluorescence to omit the dim cells. To prepare the retrovirus, pSUPER (Oligoengine, Seattle, WA), pGP (Akagi et al., 2003) and pCMV-VSV-G-RSV-Rev were co-transfected into Lenti-X 293T cells by using polyethylenimine. MDCK cells were incubated with the retrovirus for 2 days. MDCK cells stably expressing Picchu, Raichu-Ras or Raichu-Rac1 with a H-Ras membrane-targeting sequence (HRasCAAX:5′-AAGCTGAACCC-TCCTGATGAGAGTGGCCCCCGGCTGCATGAGCTGCAAGTGTGTGC-TCTCC-3′), ERK-KTR–mCherry, Rab5S34N, HRasS17N, Rac1T17N, chGrb2/iK6DHFR-iRFP713, miRFP703-iK6DHFR-SOScat, mCherry–Merlin or mCherry were established with a piggyBac transposon system. pPB plasmids [pPBbsr2-5102HRasCT(Picchu), pPBbsr2-Raichu-454HRasCT, pPBbsr2-RaichuEV-Rac1HRasCT, pPBpuro-ERK-KTR-mCherry, pPBpuro-Rab5S34N, pPBpuro-HRasS17N, pPBpuro-Rac1T17N, pPBpuro-chGrb2/iK6DHFR-iRFP713, or pPBpuro-miRFP703-iK6DHFR-SOScat, pPBneo-UbC-mCherry-Merlin, or pPBneo-UbC-mCherry] and pCMV-

mPBase(neo-) encoding *piggyBac* transposase were co-transfected into MDCK cells by electroporation with an Amaxa nucleofector (Lonza, Basel, Switzerland), followed by selection with 10 μg ml⁻¹ blasticidin S (no. 029-18701; Wako) for bsr, 2 μg ml⁻¹ puromycin (no. P-8833; Sigma-Aldrich) for puro, or 80 μg/ml G418 (no. 23985-94; Nacalai Tesque, Kyoto, Japan). After the selection, cells expressing 5102HRasCT, Raichu-454HRasCT or ERK-KTR–mCherry were subjected to single-cell cloning. For the co-expression of ERK-KTR–mCherry and H2B–iRFP, MDCK cells expressing ERK-KTR–mCherry were incubated with the lentivirus and after 2 days of incubation, the cells were treated with 100 mg ml⁻¹ zeocin (no. 11006-33-0; InvivoGen) for the selection. The obtained MDCK cells were subjected to single-cell cloning to achieve a uniform expression level of the ERK-KTR–mCherry. In some experiments, Raichu-Ras was introduced into an obtained clone co-expressing ERK-KTR–mCherry and H2B–iRFP. Merlin-KO MDCK cells (see below) expressing mCherry–Merlin or mCherry were sorted using a FACS Aria IIu cell sorter (Becton Dickinson, Franklin Lakes, NJ) with mCherry fluorescence to obtain cells with uniform expression level of mCherry–Merlin or mCherry. The cell lines used in this paper are listed in Table S1B. Note that the codon usage of Merlin was modified to minimize similarity with the gRNAs of Merlin.

## CRISPR-Cas9-mediated knockout of SOS1 and Merlin
For CRISPR/Cas9-mediated KO of dog SOS1, single guide RNAs (sgRNA) targeting the catalytic domain were designed using the CRISPRdirect (https://crispr.dbcls.jp; Naito et al., 2015). The following targeting sequences were used: SOS1#1, 5′-CAGTCGAGTGGCATATAAGC-3′; SOS1#2, 5′-GTACAACTGCGGTAAGCAGT-3′. Oligonucleotide DNAs for the sgRNA were cloned into the PX459 vectors (Ran et al., 2013), and the vectors were transfected into MDCK cells by electroporation. The transfected cells were treated with 4 mg ml⁻¹ puromycin (no. P-8833; Sigma-Aldrich) for selection. After the selection and subsequent single-cell cloning, the KO of the protein of interest in each clone was checked by immunoblotting.

Expression plasmids coding Cas9 and sgRNA for knockout of Merlin were reported previously (Boocock et al., 2021). The following sequences were used for the sgRNA sequences: 5′-CCTGGCTTCTTACGCCGTCC-3′ (sgRNA1) and 5′-GACCCCTCTGTTCACAAACG-3′ (sgRNA2). To increase the knockout efficiency, the sgRNA1 and sgRNA2 together with Cas9 were simultaneously introduced into MDCK cells by the lentivirus. The infected cells were selected with 2 μg ml⁻¹ puromycin (no. P-8833; Sigma-Aldrich) and 100 μg ml⁻¹ zeocin (no. 11006-33-0; InvivoGen). The effectiveness of these sgRNAs in reducing the expression levels of Merlin was validated by immunoblotting. Bulk cells were used for the experiments.

## shRNA-mediated KD cell line
shRNA-mediated KD of dog Rac1 was as reported previously (Hino et al., 2020). The following sequence was used for shRNA target sequence: Rac1, 5′-GCCTTCGCACTCAATGCCAAG-3′. The shRNA was introduced into MDCK cells by retroviral infection. The infected cells were selected with 4.0 mg ml⁻¹ puromycin. After the selection, reduction in expression levels of the target proteins was confirmed by immunoblotting.

## Time-lapse imaging of MDCK cells
Images of MDCK cells were collected and processed using basically the same conditions and procedures as previously described (Aoki and Matsuda, 2009). Briefly, for FRET imaging, cells were observed with IX81 and IX83 inverted microscopes (Evident, Tokyo). The IX81 inverted microscope was equipped with a UPlanAPO 10×/0.40 (Evident), a UPlanSAPO 20×/0.75 (Evident), or a UPlanSAPO 40×/0.95 objective lens (Evident), a QI-695 CCD camera (Molecular Devices, Sunnyvale, CA), a CoolLED precisExcite light-emitting diode (LED) illumination system (Molecular Devices), an IX2-ZDC laser-based autofocusing system (Evident), an MD-WELL96100T-Meta automatically programmable *xy* stage (Sigma Koki, Tokyo) and a stage top incubator (Tokai Hit, Fujinomiya, Japan). The filters and dichromatic mirrors used for time-lapse imaging with the IX81 microscope were as follows: for FRET imaging, an FF02-438/24-25 excitation filter (Semrock, Rochester, NY), an FF458-Di02-25x36 (Semrock) dichromatic mirror, and an FF01-483/32-25

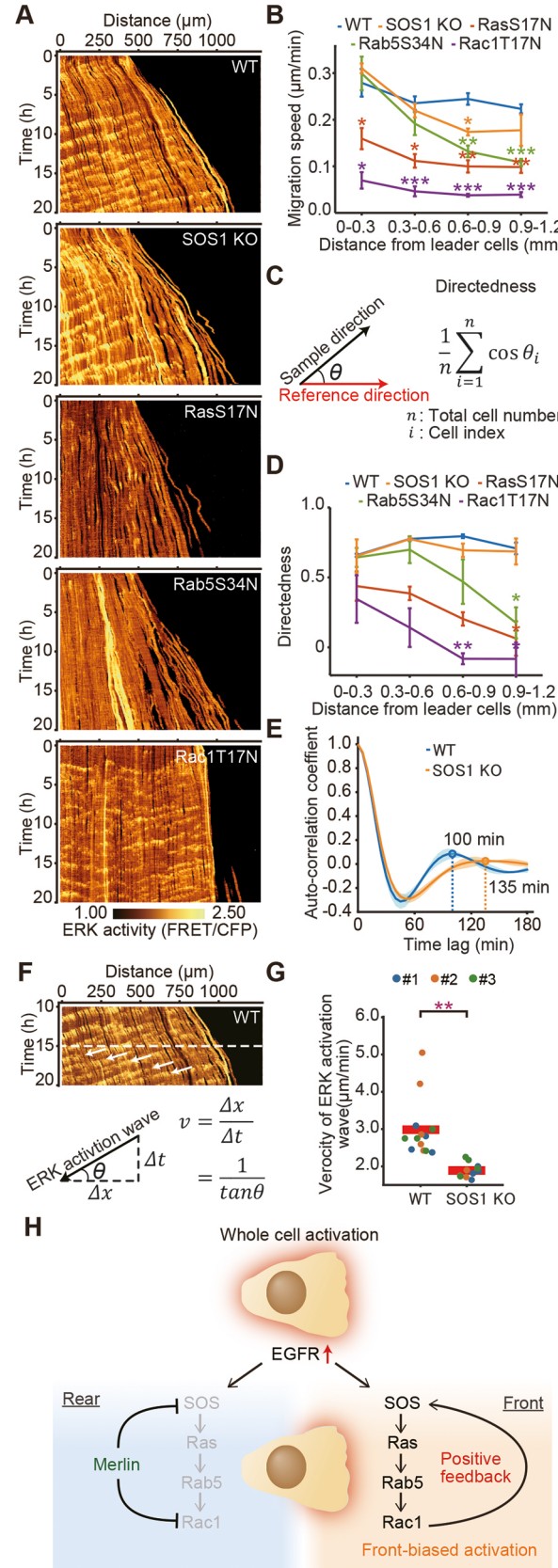

**Fig. 6. Collective cell migration is coordinated by SOS, Ras, Rac1 and Rab5 with different modes of action.** (A) Parent MDCK cells, SOS1-knockout MDCK cells or dominant-negative GTPases-expressing MDCK cells that express EKARrEV were subjected to a confinement release assay. X-T kymographs of the FRET/CFP ratio images show the ERK activity. (B) Mean±s.d. speed of cell migration after 20 h of migration binned every 300 μm from the leader cells is plotted over the distance of leader cells. The data were obtained from three independent experiments. ***$P<0.0001$; **$P<0.001$; *$P<0.01$ (unpaired two-tailed Welch's $t$-test). (C) θ is the angle between the reference direction and sample direction. Directedness was calculated by the indicated equation. The reference direction is set to the right for analysis of cell migration direction. (D) Mean±s.d. of cell migration directedness, as described in C, is plotted over the distance from the leader cells after 20 h of migration, binned every 300 μm. Data represent three independent experiments. **$P<0.001$; *$P<0.01$ (unpaired two-tailed Welch's $t$-test). (E) Temporal auto-correlations of ERK activity of the parent MDCK cells or SOS1 knockout MDCK cells located more than 0.30 mm distant from the leading edge. The bold line indicates the average temporal auto-correlation coefficients with standard deviations (shaded area). The data were obtained from the same three independent experiments as per B. $n$=2417 (control) and $n$=2277 (SOS1 KO) cells from three independent experiments. (F) A schematic of kymograph analysis. The angle of ERK activation wave was measured using ImageJ. Waves occurring between 15 and 20 h after the onset of migration were analyzed. (G) Quantified data for the ERK activation wave as shown in F. Each dot corresponds to an ERK activation wave. Data from three independent experiments are shown. Red lines represent the mean values. Results are shown for a total of $n$=13 (control) and $n$=11 (SOS1 KO) ERK activation waves. **$P<0.001$ (unpaired two-tailed Welch's $t$-test for the velocity). (H) A schematic of the positive feedback loop of small GTPases. EGFR confers temporal information without front–rear polarity. Merlin instructs cells in the front–rear polarity and regulates SOS activity.

precisExcite light-emitting diode (LED) illumination system (Molecular Devices), an IX3-ZDC2 laser-based autofocusing system (Evident), an MD-WELL96100T-Meta automatically programmable XY stage (Sigma Koki) and a stage top incubator (Tokai Hit). The filters and dichromatic mirrors used for time-lapse imaging with the IX83 inverted microscope were as follows: for FRET imaging, an ET430/ 24× excitation filter (Chroma Technology, Bellows Falls, VT), an XF2034 (455DRLP) dichromatic mirror (Omega Optical., Brattleboro, VT), and an FF01-483/32-25 (Semrock) and an ET535/30m emission filter (Chroma Technology) for CFP and FRET, respectively. For mCherry imaging, a ET572/35x excitation filter (Chroma Technology), an 89006 dichromatic mirror (Chroma Technology) and an ET632/60m (Chroma Technology) emission filter were used. For iRFP imaging, a 632/22 (Semrock) excitation filter, a FF662-FDi01 dichromatic mirror (Semrock) and a BLP01-664R (Semrock) emission filter were used.

### Confinement release assay

A confinement release assay was performed as described previously with slight modifications (Hino et al., 2020). MDCK cells were confluently seeded into a two-well culture insert (no. 81176; ibidi, Martinsried, Germany; 2.8×10⁴ cells in each well) placed on a glass-bottom dish coated with 0.3 mg ml⁻¹ type I collagen (Nitta Gelatin, Osaka, Japan). After overnight incubation, the cells were released for migration by removing the culture insert and changing the medium to Medium 199 (11043023; Life Technologies, Carlsbad, CA) supplemented with 100 units ml⁻¹ penicillin, 100 mg ml⁻¹ streptomycin and 1% bovine serum albumin (BSA) (no. A2153-50G; Sigma) or 10% fetal bovine serum (FBS) with or without additional chemicals as indicated in the figure legends.

### Ras activation by SLIPT

The self-localizing ligand-induced protein translocation (SLIPT) approach was reported previously (Suzuki et al., 2022). MDCK cells expressing Raichu-Ras with chGrb2/iK6DHFR-iRFP713 or miRFP703-iK6DHFR-SOScat (2.8×10³ cells) and MDCK cells expressing only ERK-KTR–mCherry (2.8×10⁴ cells) were mixed and seeded into the two-well culture insert placed on a glass-bottom dish coated with 0.3 mg ml⁻¹ type I

(Semrock) and FF01-542/27-25 (Semrock) emission filter for CFP and FRET, respectively. The IX83 microscope was equipped with a UPlanSAPO 20×/0.70 (Evident) or a UPlanSAPO 40×/1.4 objective lens (Evident), a Prime sCMOS camera (Photometrics, Tucson, AZ), a CoolLED

collagen. After overnight incubation, the cells were released for migration by removing the culture insert and changing the medium to Medium 199 supplemented with 100 units ml$^{-1}$ penicillin and 100 mg ml$^{-1}$ streptomycin. The cells were treated with 200 μM m$^D$cTMP to induce the translocation of chGrb2/$^{iK6}$DHFR-iRFP713 to the plasma membrane and 20 μM m$^D$cTMP to induce the translocation of miRFP703-$^{iK6}$DHFR-SOS$_{cat}$. The polarity index before the m$^D$cTMP treatment was calculated 5 min before treatment as described in the quantification of membrane-localized FRET biosensors section. The polarity index after m$^D$cTMP treatment was calculated using the averaged FRET/CFP ratio of the two time points, 0 min and 5 min after the treatment. The migration direction of each cell was determined by the cell displacement between 10 min and 5 min before treatment.

### EGF stimulation
For EGF stimulation during collective cell migration, after removing the culture insert, the medium was replaced with Medium 199 supplemented with 100 units ml$^{-1}$ penicillin, 100 μg ml$^{-1}$ streptomycin and 1% BSA. After 30 min, cells were observed under the IX83 inverted microscope. After 10 h of migration, EGF was added to 2 ng ml$^{-1}$ or 20 ng ml$^{-1}$. The Ras activation was quantified as described above in the 'Ras activation by SLIPT' section.

### Fluorescence imaging with a confocal laser microscope
Cells were observed with a Leica TCS SP8 FALCON confocal microscope (Leica-Microsystems, Wetzlar, Germany) equipped with an HC PL APO 63×/1.40 OIL CS2 objective, Lecia HyD SMD detectors, a white light laser of 80 MHz pulse frequency, a Diode 405 (VLK 0550 T01; LASOS, Jena, Germany), a 440 nm diode laser (PDL 800-D; PicoQuant, Berlin, Germany) and a stage top incubator (Tokai Hit). The following excitation wavelengths and emission band paths were used for the imaging: for CFP and FRET imaging, 440 nm excitation, 460–510 nm and 515–567 nm emission, respectively; for iRFP imaging, 670 nm excitation, 700–772 nm emission. For 3D fluorescence imaging of the plasma membrane, MDCK cells expressing Raichu-Ras were subjected to confinement release assay as described above in the plasma membrane, MDCK cells expressing Raichu-Ras were subjected to confinement. The Confinement release assay was as described in the above section. Cells were observed 10 h after migration. For the confirmation of Ras activation by SLIPT, MDCK cells expressing Raichu-Ras with chGrb2/$^{iK6}$DHFR-iRFP713 or miRFP703-$^{iK6}$DHFR-SOScat were seeded on a glass-bottom dish coated with 0.3 mg ml$^{-1}$ type I collagen. After overnight incubation, the medium was replaced with Medium 199 supplemented with 100 units ml$^{-1}$ penicillin and 100 μg ml$^{-1}$ streptomycin. After 2 h, the cells were treated with 2 μM m$^D$cTMP to induce the translocation of chGrb2/$^{iK6}$DHFR-iRFP713 or miRFP703-$^{iK6}$DHFR-SOScat.

### Image processing for the FRET/CFP ratio
Image processing for FRET/CFP ratio images was performed with Fiji. Images were processed with a median filter to reduce noise, and then the background intensity was subtracted by using the subtract-background function except the experiments in Figs 1A,B, 3A and 4A. In the experiments, a control dish filled with Medium 199 supplemented with 100 units ml$^{-1}$ penicillin, 100 μg ml$^{-1}$ streptomycin and 10% FBS was imaged as the background. The processed images were subjected to image calculation and the ratio values were binned into eight steps to obtain 8-color FRET/CFP ratio images. To convey the brightness of the original images to the FRET/CFP ratio images, the 8-color FRET/CFP ratio images were multiplied by the corresponding intensity-normalized grayscale image. For the 8-bit pseudocolor FRET/CFP ratio images, FRET images were subjected to thresholding with the Huang method (Huang and Wang, 1995) by Fiji to obtain binary images of the nucleus. Then, ratio images without binning of the ratio values were multiplied by the binary images of the nucleus to remove signals in the cell-free regions. To generate kymographs, the CFP or FRET intensity was averaged along the y-axis in a defined region of the images, providing an intensity line along the x-axis. Subsequently, the generated kymographs of the FRET images were divided by that of the CFP to obtain the kymographs of the FRET/CFP ratio. To remove the signals in the cell-free region from the kymographs of FRET/CFP, the kymographs of FRET images

were subjected to manual thresholding by Fiji and multiplied by the kymographs of FRET/CFP ratio. To calculate the velocity of ERK activation wave, the angle of ERK wave was measured using ImageJ.

### Quantification of membrane-localized FRET biosensors
For the live-cell imaging of EGFR, Ras, Rac and Rab5 activities in each cell, cells expressing membrane-localized FRET biosensors and cells expressing ERK-KTR–mCherry were seeded into a two-well culture insert at a ratio of 1:10 (3.0×10$^4$ cells in total) to prepare mosaic monolayers. The cells were imaged as described above in the 'Confinement release assay' section.

The calculation of the polarity index was performed at the 10-h timepoint of migration through the following procedures. Cellpose, an algorithm for cellular segmentation to obtain segmented cell images (Stringer et al., 2021) was applied to the FRET fluorescence image captured between 9 h 40 min and 10 h 20 min of migration. Exclusion criteria were established as follows: if Cellpose identified one cell as two or more cells, recognized more than two cells as one cell, or failed to detect a cell for more than two frames, the corresponding cell clusters were excluded from subsequent analysis. For determining the migration direction of each cell, the FIJI TrackMate plug-in (Tinevez et al., 2017) was applied to the segmented cells. FRET images underwent thresholding using the Huang method in Fiji to generate binary membrane images. The segmented cell images, migration direction, and binarized images were further processed by using MATLAB scripts (available upon request). To define the front or rear area of each cell, segmented cells were divided into two parts perpendicular to the migration direction of each cell at the center of mass of the segmented cell. Binary images were morphologically eroded with a radius of 8 pixels to emphasize the peripheral areas of cell clusters. Within a cell cluster, the total FRET or CFP intensity of pixels present in both the peripheral and front regions was calculated and named FRET$_{Front}$ or CFP$_{Front}$. The total FRET or CFP intensity in both the peripheral and rear regions was also calculated and named FRET$_{Rear}$ or CFP$_{Rear}$. Then, the polarity index was obtained by the following formula. After quantification of the polarity index, cell clusters were categorized into ERK wave (+) or ERK wave (−) clusters by the formula below:

$$\frac{FRET_{Front}/CFP_{Front}}{FRET_{Rear}/CFP_{Rear}} - 1$$

The activity of EGFR, Ras, Rac and Rab5 of the cell cluster was determined by dividing the total FRET intensity of the binary membrane image by the corresponding CFP intensity.

Cell clusters were excluded if both the polarity index and activity were identified as outliers using the Hampel identifier method. If the FRET biosensor aggregated in a cell, the corresponding cell cluster was excluded from the analysis. If the cells were not single-cloned or sorted by FACS to omit the dim cell clusters, then cell clusters were excluded based on thresholding the mean CFP intensity.

### Categorization of cell clusters into ERK wave (+) or ERK wave (−)
After calculating the polarity index, we classified each cell cluster as either ERK wave (+) or ERK wave (−), based on the ERK activity of its neighboring cells. Here, neighboring cells were defined as those directly adjacent to the periphery of the cluster. To quantify ERK activity in these cells, rectangular regions of interest (ROIs) were manually defined in the cytoplasm and nucleus of each cell. ERK activity was calculated as the ratio of the mean cytoplasmic to nuclear fluorescence intensity of ERK-KTR–mCherry. Cells undergoing mitosis were excluded from the analysis. For each cluster, the average ERK activity of the neighboring cells was calculated. Based on this value, the cluster was categorized as ERK wave (+) or ERK wave (−) by applying a threshold defined as the mean ERK activity of all clusters under the same medium condition. The threshold was set to 1.23 for medium conditions with 10% FBS and 1.10 for those without FBS.

### Immunoblotting
For immunoblotting of cell lysates, 5.0×10$^5$ MDCK cells were plated in a six-well plate (no. 140675; Thermo Fisher Scientific). At 1 day after seeding, cells were lysed with SDS sample buffer containing 62.5 mM Tris-HCl pH 6.8,

12% glycerol, 2% SDS, 40 ng ml$^{-1}$ Bromophenol Blue and 5% 2-mercaptoethanol, followed by sonication with a Bioruptor UCD-200 (Cosmo Bio, Tokyo, Japan). After boiling at 95°C for 5 min, the samples were separated by SDS-PAGE on SuperSep Ace 5–20% precast gels (Wako) and transferred to polyvinylidene difluoride membranes (Merck Millipore, Burlington, MA) for immunoblotting. Primary and secondary antibodies were diluted in Odyssey blocking buffer (LI-COR Biosciences). Proteins were detected by an Odyssey Infrared Imaging System (LI-COR Biosciences).

## Angle analysis

The experiment and the quantification of the migration direction of each cell as shown in Fig. 4E was conducted as described above in the 'Quantification of membrane-localized FRET biosensors' section. The directionality of collective cell migration was classified as right, left, up or down based on which edge of the rectangular cell sheet was observed. Subsequently, the angles between the direction of individual cell movement and the direction of collective cell migration were calculated by using MATLAB.

## Quantification of migration speed and directionality

The location of MDCK cells expressing ERK biosensors were determined by using the FIJI TrackMate plug-in (Tinevez et al., 2017). The speed of cell migration was calculated by mean displacement during the 25 min. For quantification of directionality of cell migration, we defined directedness as shown in Fig. 6C. The direction of cell migration for 20 min was defined as the sample direction of cell migration for individual cells. The reference direction was classified as right, left, up or down based on which edge of the rectangular cell sheet was observed.

## Quantification of auto-correlation coefficient and cross-correlation coefficient

Similar to the quantification of membrane-localized FRET biosensors section or the quantification of migration speed and directionality section, EKARrEV-NLS, Raichu-Ras or ERK-KTR–mCherry was quantified. For auto-correlation analysis, temporal auto-correlation of ERK activities of cells located more than 0.3 mm distant from leader cells were calculated by using MATLAB scripts. For cross-correlation analysis, temporal cross-correlation of ERK activities and Ras activities were calculated by MATLAB scripts.

## Statistical analysis

Probability (*P*) values were determined by using the T.TEST function of Microsoft Excel with two-tailed distribution and two-sample unequal variance, or a paired *t*-test. The sample number for this calculation (*n*) is indicated in each figure legend.

## Acknowledgements

We are grateful to the members of the Matsuda Laboratory for their helpful input, to K. Hirano, T. Uesugi and K. Takakura, who provided technical assistance, and to the Medical Research Support Center of Kyoto University for DNA sequence analysis.

## Competing interests

The authors declare no competing or financial interests.

## Author contributions

Conceptualization: Y.J., M.M., K.T.; Data curation: Y.J., M.M., K.T.; Formal analysis: Y.J., M.M., K.T.; Funding acquisition: M.M., K.T.; Investigation: Y.J.; Methodology: Y.J., M.M., K.T.; Project administration: M.M., K.T.; Resources: E.D., K.M., N.H., S.T., M.M., K.T.; Supervision: M.M., K.T.; Validation: Y.J., M.M., K.T.; Writing – original draft: Y.J.; Writing – review & editing: Y.J., M.M., K.T.

## Funding

This work was supported by the Kyoto University Live Imaging Center. Financial support was provided by Japan Society for the Promotion of Science (JSPS) KAKENHI grants (21H05226 to K.T., 19H00993 and 20H05898 to M.M.), a Japan Science and Technology Agency (JST) CREST grant (JPMJCR1654 to M.M.), and a JST Moonshot Research and Development Program grant (JPMJPS2022 to M.M.). Open Access funding provided by Tokushima University. Deposited in PMC for immediate release.

## Data and resource availability

All relevant data can be found within the article and its supplementary information.

## Peer review history

The peer review history is available online at https://journals.biologists.com/jcs/lookup/doi/10.1242/jcs.263779.reviewer-comments.pdf

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
