## [Peer Review File · Journal of Cell Science]

Front-biased activation of the Ras-Rab5-Rac1 loop coordinates collective cell migration.

Yuya Jikko, Eriko Deguchi, Kimiya Matsuda, Naoya Hino, Shinya Tsukiji, Michiyuki Matsuda and Kenta Terai

DOI: 10.1242/jcs.263779

Editor: Guillaume Jacquemet

Review timeline

Original submission:	5 December 2024
Editorial decision:	6 January 2025
First revision received:	10 April 2025
Editorial decision:	7 May 2025
Second revision received:	2 July 2025
Accepted:	10 July 2025

Original submission

First decision letter

MS ID#: jcs.263779

MS TITLE: Collective cell migration is coordinated by front-biased activation of the positive feedback loop consisting of Ras, Rab5, and Rac1.

AUTHORS: Yuya Jikko; Eriko Deguchi; Kimiya Matsuda; Naoya Hino; Shinya Tsukiji; Michiyuki Matsuda; Kenta Terai

ARTICLE TYPE: Research Article

Dear Dr Terai,

We have now reached a decision on the above manuscript.

To see the reviewers' reports and a copy of this decision letter, please go to:

Reviewer 1

Advance summary and potential significance to field

In this work, collective cell migration is under scrutiny. Collective cell migration is shown to be orchestrated by the front-to-rear propagation of EGFR-Ras-ERK pathway activation. However, the mechanisms that integrate this intercellular signaling with cellular front-side specification had remained elusive. Recent research utilizing FRET biosensors and chemogenetic tools has provided insights into this process. In this work, these authors provide clear evidence that while EGFR activation occurs uniformly across cells, Ras activation exhibits a front-biased distribution, dependent on the tumor suppressor protein Merlin and the small GTPase Rac1, both of which also show front-biased activity. Additionally, Rab5, a key regulator of endocytosis and cell migration, displays similar front-biased activation, functioning downstream of Ras and being essential for Rac1 activation in line with previous literature. These findings suggest a positive feedback loop involving Ras, Rab5, and Rac1 that is predominantly active at the leading edge of migrating cell collectives,

offering new spatiotemporal understanding of front-rear information processing during collective migration.

Comments for the author

The study is meticulously conducted and the interpretations are robust. The manuscript is a combination of elegant experiments, which are rigorously interpreted providing sufficient proof of a key pathway potentially driving collective motion not only in MDCK but more generally, albeit no other cell types beside MDCK is analyzed.

It would be beneficial to assess whether Merlin exhibits a planar polarized distribution along the ERK activation wave as this work and previous studies have indicated. Given the methodologies employed, this analysis appears straightforward and could provide further insights into the spatial dynamics of Merlin in relation to ERK signaling during collective cell migration. Also is merlin silencing impacting on collective motion?

Additionally, when assessing the impact of SOS1 into the pathways it is unclear whether SOS1 silencing, which is shown to reduce the front-to-back polarization of RAS activity has any impact on actual wound speed and cell locomotion as one would expect.

Reviewer 2

Advance summary and potential significance to field

This manuscript by Jikko et al. investigates the activity of different proteins in the Ras-Erk signaling cascade during the collective cell migration of MDCK cells. They use an array of powerful and elegant tools, including FRET probes and chemogenomic constructs, to determine the relationship between the different proteins of the cascade (EGFR, the RasGEF SOS, Ras, Rac1, Rab5), their polarization and their response to ERK activity waves. They found that all but EGFR were biased with a slightly higher activity at the front and functionally interconnected. This may provide a positive feedback loop in the front of the cells with SOS activating Ras, which in turn would act on Rab5, Rab5 acts on Rac1, and then Rac1 reinforces SOS. Whereas at the rear of the cluster, Merlin would inhibit Rac1 and SOS.

Overall, the study is well performed, with elegant experiments. However, I have several concerns that will be developed thereafter: 1) the polarization of the different probes is very subtle, 2) some experiments are not well explained or difficult to interpret, 3) the impact of the positive feedback loop on the migration process is not addressed, 4) hence, as a whole, the significance of the data is unclear, 5) I found the manuscript very complicated to read, and it would benefit from a real discussion after the result section.

Comments for the author

Major Concerns:

1) My understanding is that the polarization of the activity of Ras, Rac1, Rab5 is at maxima 5% more at the front than at the back. This is calculated through a ratio of activities, themselves calculated by ratios (activity front/activity back -1). Activity is the ratio of FRET/CFP. A couple of thoughts: First, using ratios in statistics can be misleading as it may artificially increase values, especially when the divider is small. It seems not the case here based on the activity values that are presented, but it is still a concern to me, although I have no good alternative to propose. More important, I understand that values are statistically significant, but are they biologically significant. Is there any data that suggests that such a small difference of signal between the front and the back regulates directionality, persistency or migration speed?

2) I am still confused about how ERK wave (+) and (-) are defined. I understand that it is dependent on the ERK activity in the surrounding cells, but, by eyes, it seems that in most cases the surrounding cells are not homogeneously ERK (+) or (-), if not different from what is stated. For example, Fig. 2A, ERK-KTR in the EGF(-) wave (-) panel looks similar to both wave (+) panels. Fig. 2I, in the SOS KO, the wave (+) and (-) panels look similar. In Fig. 3A, in the control panels, the top left cells seem to have more nuclear ERK in the wave (+) than in the wave (-), while the lower cells have indeed more nuclear ERK in the wave (-). The Rac1T17N wave (-) panel contains both cells with nuclear ERK and little to no nuclear ERK, etc. Overall, this heterogeneity seems to be problematic as the cells are not in "black or white" situations where they are entirely in a ERK wave (-) or (+) situation, which is concerning as the cells that are observed per se might be even different than the surrounding cells. Finally, there seems to be no quantitative estimation of the level of ERK signal surrounding the cells.

Furthermore, the kymographs shown in Fig. 1, 4 and 5 are not well explained. From my reading of the other publications of this group concerning ERK signaling, my understanding is that what is important is the periodicity of the activation of the different proteins measured by FRET. This should be better described. I am also wondering if the authors could determine the rhythm of the activation and show that the rhythms are similar to the other probes (which it seems by eyes) and to ERK. This would be a strong indication that these proteins are in the same pathway and activated sequentially.

Finally, I was expecting to see a vesicular Rab5 signal and not a cortical one. Can the authors comment on that and do they know that the probe is properly localized and indeed represent Rab5a activity in their cells.

3) The authors found that there is more activity of SOS, Ras, Rac1 and Rab5 at the front and propose a positive feedback loop in the ERK wave (+), however the role of this process in migration is unclear. The author found that in SOS KO cells do not polarize Ras activity depending on the ERK wave. Are SOS1 KO cells migrating normally? Also, how to explain that the use of SOS catalytic domain does not abolish the polarization of Ras in ERK wave (+)? This was unclear to me. Overall, a limitation of this study is that the authors basically never determine the impact of their findings on cell migration.

4) The fact that all the phenotype observed are subtle and that their impact on migration is not tested limits significantly the scope of this study. The authors have different tools that could be exploited to perturb the Ras-Rac-Rab5 cascade and test the impact on migration.

5) I found that the manuscript is difficult to read. I provide some examples in the "minor concerns" (see below). Overall, I found that the experiments are not explained with sufficient details. Some figures are not self-explanatory (e.g. kymographs). The rationale for some experiments is not obvious at first read. For example, the use of the chemogenomic tools is very elegant, but the description of the experiment is not straightforward. I would suggest that the authors first state the question to be tested before providing the experimental set up. For example, L.156 could be modified in something like: To examine whether SOS regulation is involved in the front-biased activation, we targeted SOS and the catalytic domain of SOS, which does not contain the regulatory domains, at the plasma membrane. For this we employed the self-localizing...).

Finally, I would strongly recommend the addition of a discussion section to recapitulate the data, explain the importance of the study and elaborate on its significance.

Minor Concerns:

1) The authors should clearly state in the abstract, in the introduction and in the result section that their subject of study is MDCK cells (or at least of epithelial cells). To my knowledge ERK waves and the signaling cascade characterized here are not universally employed for collective cell migrations.

2) The summary mention front-side specification, but the manuscript does not address the distribution on "sides".

3) L.71-72, I do not understand the rationale for this affirmation. Maybe a source is lacking.

4) L. 74, Rab should be Rab5

- 5) L.84-86, the mechanism of the EGFR FRET probes (and other probes thereafter) could be supported by a scheme in the figure (or a supplemental figure describing all the probes?). This would be helpful for the reader.
- 6) There is a typo in Fig. 1G (magnified).
- 7) L.126 Border cell migration is driven both by PVR and EGFR, PVR seems to be dominant for most of the migration process and tools to study RTK activity in border cells do not discriminate between PVR and EGFR.
- 8) L. 133, the suggestion that mechanical forces are also involved in the process is mentioned there and also in the introduction, but this is neither tested nor integrated in the model and as such I would move this in a discussion section where I would properly address this question.
- 9) L.138 Movie 2 is missing after Fig.1M.
- 10) L.138, it may be important to clearly state that the same cell cluster is analyzed in wave (-) and (+) conditions.
- 11) Ideally, a catalytically inactive SOS should be used as a control for Fig.2.
- 12) L.177, "suppressed" should be replaced by "reduced" as there is still significant Ras activity.
- 13) L.178, "This partial effect could be attributed to the presence of SOS2". I would state the speculative nature of this sentence more explicitly.
- 14) L.220, I don't think that these data indicate that Rac1 functions downstream of Rab5, but that Rab5's activity is required for proper activation of Rac1.
- 15) Similarly, l.226 seems to be overly conclusive, especially in the context of a positive feedback loop, since then every protein is upstream and downstream of other constituents of the loop.

First revision

Author response to reviewers' comments

Reviewer 1: SUMMARY OF THE ADVANCE MADE IN THIS PAPER AND ITS POTENTIAL SIGNIFICANCE TO THE FIELD

In this work, collective cell migration is under scrutiny. Collective cell migration is shown to be orchestrated by the front-to-rear propagation of EGFR-Ras-ERK pathway activation. However, the mechanisms that integrate this intercellular signaling with cellular front-side specification had remained elusive. Recent research utilizing FRET biosensors and chemogenetic tools has provided insights into this process. In this work, these author provide clear evidence that while EGFR activation occurs uniformly across cells, Ras activation exhibits a front-biased distribution, dependent on the tumor suppressor protein Merlin and the small GTPase Rac1, both of which also show front-biased activity. Additionally, Rab5, a key regulator of endocytosis and cell migration, displays similar front-biased activation, functioning downstream of Ras and being essential for Rac1 activation in line with previous literature. These findings suggest a positive feedback loop involving Ras, Rab5, and Rac1 that is predominantly active at the leading edge of migrating cell collectives, offering new spatiotemporal understanding of front-rear information processing during collective migration.

SUGGESTIONS TO AUTHORS

The study is meticulously conducted and the interpretations are robust. The manuscript is a combination of elegant experiments, which are rigorous interpreted providing sufficient proof of a key pathays potentially driving collective motion not only in MDCK but more generally, albeit no other cell types beside MDCK is analyzed.

It would be beneficial to assess whether Merlin exhibits a planar polarized distribution along the ERK activation wave as the this work and previous studies have indicated. Given the methodologies employed, this analysis appears straightforward and could provide further insights into the spatial dynamics of Merlin in relation to ERK signaling during collective cell migration. Also is merlin silencing impacting on collective motion?

We really appreciate the supportive comments by the reviewer. We tried several times to visualize the translocation of GFP-tagged Merlin. Disappointingly, we were not able to observe the translocation even with super-resolution microscopy. We assumed that the amount of translocated GFP is below the detection level. Definitely, further investigation and improvement of the microscopy are required.

We have modified the text to mention the above issue (line 200).

“We also attempted to visualize the translocation of Merlin tagged with EGFP during collective cell migration. However, the attempt was unsuccessful, possibly because the EGFP tag interfered with the proper localization or dynamics of Merlin during collective migration. Additionally, the spatiotemporal regulation of Merlin may be too subtle or transient to be captured under our imaging conditions.”

Additionally, when assessing the impact of SOS1 into the pathways it is unclear whether SOS1 silencing, which is shown to reduced the front-to-back polarization of RAS activity has any impact on actual wound speed and cell locomotion as one would expect.

According to the reviewer’s comments, we re-analyzed the impact of the SOS1 knockout and other molecules on collective cell migration. Interestingly, a role of SOS1 is a rhythm coordinator. Depletion of SOS1 slowed the ERK wave rhythm, as shown in Fig. 6. At this point, we cannot explain why SOS1 depletion led to the observed phenotype during collective cell migration, but we speculate that disrupting the rhythm of the ERK wave may be involved. We have modified the text to mention the above issue (line 250).

Reviewer 2: SUMMARY OF THE ADVANCE MADE IN THIS PAPER AND ITS POTENTIAL SIGNIFICANCE TO THE FIELD

This manuscript by Jikko et al. investigates the activity of different proteins in the Ras-Erk signaling cascade during the collective cell migration of MDCK cells. They use an array of powerful and elegant tools, including FRET probes and chemogenomic constructs, to determine the relationship between the different proteins of the cascade (EGFR, the RasGEF SOS, Ras, Rac1, Rab5), their polarization and their response to ERK activity waves. They found that all but EGFR were biased with a slightly higher activity at the front and functionally interconnected. This may provide a positive feedback loop in the front of the cells with SOS activating Ras, which in turn would act on Rab5, Rab5 acts on Rac1, and then Rac1 reinforce SOS. Whereas at the rear of the cluster, Merlin would inhibit Rac1 and SOS.

Overall, the study is well performed, with elegant experiments. However, I have a several concerns that will be developed thereafter: 1) the polarization of the different probes is very subtle, 2) some experiments are not well explained or difficult to interpret, 3) the impact of the positive feedback loop on the migration process is not addressed, 4) hence, as a whole, the significance of the data is unclear, 5) I found the manuscript very complicated to read, and it would benefit from a real discussion after the result section.

SUGGESTIONS TO AUTHORS

Major Concerns:

1) *My understanding is that the polarization of the activity of Ras, Rac1, Rab5 is at maxima 5% more at the front than at the back. This is calculated through a ratio of activities, themselves calculated by ratios (activity front/activity back -1). Activity is the ratio of FRET/CFP. A couple of thoughts: First, using ratios in statistics can be misleading as it may artificially increase values, especially when the divider is small. It seems not the case here based on the activity values that are presented, but it is still a concern to me, although I have no good alternative to propose. More important, I understand that values are statistically significant, but are they biologically significant. Is there any data that suggests that such a small difference of signal between the front and the back regulates directionality, persistency or migration speed?*

Thank you for pointing this out. We agree with your concern and have performed additional experiments. To investigate the correlation between polarity and correct cell migration, we inhibited the activation of each molecule through knockout or dominant-negative expression. Ras polarity was disturbed by inhibiting Rac1 activation (Fig. 3B) and depleting SOS1 (Fig. 2J). Under these conditions, the migration speed of follower cells was retarded, as shown in the new figure, Fig. 6B. The expression of the dominant-negative mutant of Rab5 disturbed the Rac1 polarity (Fig. 4D) and follower cells migration (Fig. 6D), supporting that the values are biologically significant. Of course, we are still wondering how the cell recognizes such a tiny gradient of activity. Further investigation and improvement of the biosensors are required. We have modified the text to mention the above issue (line 250).

2) *I am still confused about how ERK wave (+) and (-) are defined. I understand that it is dependent on the ERK activity in the surrounding cells, but, by eyes, it seems that in most cases the surrounding cells are not homogeneously ERK (+) or (-), if not different from what is stated. For example, Fig. 2A, ERK-KTR in the EGF(-) wave (-) panel looks similar to both wave (+) panels. Fig. 2I, in the SOS KO, the wave (+) and (-) panels look similar. In Fig. 3A, in the control panels, the top left cells seem to have more nuclear ERK in the wave (+) than in the wave (-), while the lower cells have indeed more nuclear ERK in the wave (-). The Rac1T17N wave (-) panel contains both cells with nuclear ERK and little to no nuclear ERK, etc. Overall, this heterogeneity seems to be problematic as the cells are not in "black or white" situations where they are entirely in a ERK wave (-) or (+) situation, which is concerning as the cells that are observed per se might be even different than the surrounding cells. Finally, there seems to be no quantitative estimation of the level of ERK signal surrounding the cells.*

We appreciate the reviewer's comments. We have explained the details of the procedure as thoroughly as possible and shown them in Sup Fig 1C and 1D (line 673). Briefly, we selected the neighboring cells of the cluster, measured the nuclear and cytoplasmic intensity of the cells, and designated the ERK wave-positive clusters.

*Furthermore, the kymographs shown in Fig. 1, 4 and 5 are not well explained. From my reading of the other publications of this group concerning ERK signaling, my understanding is that what is important is the periodicity of the activation of the different proteins measured by FRET. This should be better described. I am also wondering if the authors could determine **the rhythm of the activation** and show that the rhythms are similar to the other probes (which it seems by eyes) and*

to ERK. This would be a strong indication that these proteins are in the same pathway and activated sequentially.

As anticipated, the synchronized activation rhythm between ERK and Ras, as well as ERK and EGFR, is shown in Sup Fig 1A and 1B, supporting our model. Meanwhile, the biased activity, as indicated by the polarity index in Fig 4D and Fig 5D, was exaggerated only in the ERK wave-positive clusters, demonstrating the synchrony of the activation rhythm between ERK and Rac1, as well as ERK and Rab5.

We have modified the text and mentioned the issue (line 273).

“Supporting this, the synchronized activation rhythm between ERK and Ras, as well as ERK and EGFR, is shown in SupFig 1A and 1B. Although ERK wave propagation was unclear in the kymographs of Raichu-Rac1 and Raichu-Rab5 expressing cells, the biased activity, as indicated by the polarity index in Fig 4D and Fig 5D, was exaggerated only in the ERK wave-positive clusters, demonstrating the synchrony of the activation rhythm between ERK and Rac1, as well as ERK and Rab5. ”

Finally, I was expecting to see a vesicular Rab5 signal and not a cortical one. Can the authors comment on that and do they know that the probe is properly localized and indeed represent Rab5a activity in their cells.

In our assay, we focused on Rab5 activity just below the plasma membrane by fusing a membrane-anchoring signal peptide. That would be the reason why the vesicles were missing.

We have modified the text and mentioned the issue (line 233).

“We extended our visualization to Rab5 by using the Rab5 biosensor, Raichu-Rab5 (Supplementary Figure 1E), which is anchored to the plasma membrane to measure activity just beneath the membrane and exclude vesicular signals.”

3) The authors found that there is more activity of SOS, Ras, Rac1 and Rab5 at the front and propose a positive feedback loop in the ERK wave (+), however the role of this process in migration is unclear. The author found that in SOS KO cells do not polarize Ras activity depending on the ERK wave. Are SOS1 KO cells migrating normally? Also, how to explain that the use of SOS catalytic domain does not abolish the polarization of Ras in ERK wave (+)? This was unclear to me. Overall, a limitation of this study is that the authors basically never determine the impact of their findings on cell migration.

4) The fact that all the phenotype observed are subtle and that their impact on migration is not tested limits significantly the scope of this study. The authors have different tools that could be exploited to perturb the Ras-Rac-Rab5 cascade and test the impact on migration.

According to the reviewer’s comments, we re-analyzed the impact of SOS1 knockout and other molecules on collective cell migration, as shown in Fig. 6. Interestingly, a role of SOS1 is a rhythm coordinator. Depletion of SOS1 slowed the ERK wave rhythm, as shown in Fig. 6. At this point, we cannot explain why SOS1 depletion led to the observed phenotype during collective cell migration, but we speculate that disrupting the rhythm of the ERK wave may be involved (line 250).

In the case of other molecules, such as Ras, Rac1, and Rab5, they also demonstrated slower migration speeds in follower cells, supporting the idea that these molecules are part of the same pathway. Note that collectively migrating cells consist of two different types: leader cells and follower cells. The former migrates toward free space. Thus, the results of Fig. 6 also suggest that Ras and Rac activation is essential for leader cell migration, as we have reported.

5) I found that the manuscript is difficult to read. I provide some examples in the "minor concerns" (see below). Overall, I found that the experiments are not explained with sufficient details. Some figures are not self-explanatory (e.g. kymographs). The rationale for some experiments is not obvious at first read. For example, the use of the chemogenomic tools is very elegant, but the description of the experiment is not straightforward. I would suggest that the authors first state the question to be tested before providing the experimental set up. For example, l.156 could be modified in something like: *To examine whether SOS regulation is involved in the front-biased activation, we targeted SOS and the catalytical domain of SOS, which does not contain the regulatory domains, at the plasma membrane. For this we employed the self-localizing...*

We appreciate the positive comments. We have modified the text (line 160).

"Thus, we hypothesized that the SOS1-biased translocation causes the front-biased activation. To test this model, the self-localizing ligand-induced protein translocation (SLIPT) assay (31) to eliminate the biased translocation."

Finally, I would strongly recommend the addition of a discussion section to recapitulate the data, explain the importance of the study and elaborate on its significance.

We have added the following sentence (line271).

"We have demonstrated that the front-side biased activation of Ras, Rab5, and Rac1, but not EGFR, occurs during collective migration in MDCK cells, primarily caused by SOS1-biased activation. Supporting this, the synchronized activation rhythm between ERK and Ras, as well as ERK and EGFR, is shown in SupFig 1A and 1B. Although ERK wave propagation was unclear in the kymographs of Raichu-Rac1 and Raichu-Rab5 expressing cells, the biased activity, as indicated by the polarity index in Fig 4D and Fig 5D, was exaggerated only in the ERK wave-positive clusters, demonstrating the synchrony of the activation rhythm between ERK and Rac1, as well as ERK and Rab5."

Minor Concerns:

1) *The authors should clearly state in the abstract, in the introduction and in the result section that their subject of study is MDCK cells (or at least of epithelial cells). To my knowledge ERK waves and the signaling cascade characterized here are not universally employed for collective cell migrations.*

We have emphasize the aim of our study (line 66).

"The aim of this study is to elucidate the molecular mechanisms that coordinate the front-to-rear intercellular propagation of EGFR-Ras-ERK pathway activation during collective cell migration, particularly identifying the determinants of cellular front-side specification. To address this, we visualized the activity of EGFR, Ras, Rac1, and Rab5 using FRET biosensors in Madin-Darby Canine Kidney (MDCK) cells. "

2) *The summary mention front-side specification, but the manuscript does not address the distribution on "sides".*

We have amended as below,

"We have demonstrated that the front-side biased activation of Ras, Rab5, and Rac1, but not EGFR, occurs during collective migration in MDCK cells, primarily caused by SOS1-biased activation."

3) *L.71-72, I do not understand the rationale for this affirmation. Maybe a source is lacking.*

We have amended as below,

“These observations suggest that EGFR ligands, which are chemical signals, only convey information about timing, while the information about the front and back of the cell is controlled by Rac1”

4) L. 74, *Rab* should be *Rab5*

We have made amendments based on the comments.

5) L.84-86, *the mechanism of the EGFR FRET probes (and other probes thereafter) could be supported by a scheme in the figure (or a supplemental figure describing all the probes?). This would be helpful for the reader.*

We have added the schemes as SupFig1

6) *There is a typo in Fig. 1G (magnified).*

We have made amendments.

7) L.126 *Border cell migration is driven both by PVR and EGFR, PVR seems to be dominant for most of the migration process and tools to study RTK activity in border cells do not discriminate between PVR and EGFR.*

In this study, we have focused on the follower cells. We have also emphasized the definitions of 'follower' and 'leader,' or 'border,' in the text (line 253).

“In this assay system, cells located 0-0.3 mm from the edge were regarded as leader cells, while those at 0.6-1.2 mm were classified as follower cells. ”

8) L. 133, *the suggestion that mechanical forces are also involved in the process is mentioned there and also in the introduction, but this is neither tested nor integrated in the model and as such I would move this in a discussion section where I would properly address this question.*

We agree with the comment. However, in this study, we do not assess the involvement of mechanical forces; therefore, we have refrained from discussing this issue due to the lack of experimental results.

9) L.138 *Movie 2 is missing after Fig.1M.*

10) L.138, *it may be important to clearly state that the same cell cluster is analyzed in wave (-) and (+) conditions.*

We have amended the text.

11) *Ideally, a catalytically inactive SOS should be used as a control for Fig.2.*

We assume the reviewer is concerned about any side effects of the TMP treatment. As far as we have tested, we could not detect any cross-activation of other pathways (PMID: 39546398, 36174555). Additionally, the catalytically inactive mutant functions as a dominant-negative form (PMID: 9047390). Therefore, we could not address the comment.

- 12) L.177, "suppressed" should be replaced by "reduced" as there is still significant Ras activity.
13) L.178, "This partial effect could be attributed to the presence of SOS2". I would state the speculative nature of this sentence more explicitly.
14) L.220, I don't think that these data indicate that Rac1 functions downstream of Rab5, but that Rab5's activity is required for proper activation of Rac1.
15) Similarly, l.226 seems to be overly conclusive, especially in the context of a positive feedback loop, since then every protein is upstream and downstream of other constituents of the loop.

We have amended the text.

Second decision letter

MS ID#: jcs.263779R1

MS TITLE: Front-biased activation of Ras-Rab5-Rac1 loop coordinates collective cell migration.

AUTHORS: Yuya Jikko; Eriko Deguchi; Kimiya Matsuda; Naoya Hino; Shinya Tsukiji; Michiyuki Matsuda; Kenta Terai

ARTICLE TYPE: Research Article

Dear Dr Terai,

We have now reached a decision on the above manuscript.

To see the reviewers' reports and a copy of this decision letter, please go to:

As you will see, the reviewers gave favourable reports but raised one critical point that will require amendments to your manuscript (with textual changes). I hope that you will be able to carry these out because I would like to be able to accept your paper, depending on further comments from reviewers.

Reviewer 1

Advance summary and potential significance to field

I commend the authors for re-analyzing their data in response to the comment and for identifying a novel and intriguing role of SOS1 as a potential regulator of ERK wave rhythm. This is a valuable addition to the understanding of collective migration dynamics and suggests an interesting layer of signaling regulation.

Yet the impact of SOS1 depletion on collective migration based on the metric used, appear based on the metric used marginal albeit significant. I do understand that the major focus is on the migration of the follower,

This said, the idea that SOS1 affects the rhythm of ERK waves is intriguing, it would be helpful to comment/speculate how this alteration translates into measurable changes in migration efficiency, or whether the impact is more subtle. If the latter, a brief discussion highlighting this nuance would improve the mechanistic clarity of the manuscript and help readers appreciate the functional relevance of ERK wave rhythm in this context.

Reviewer 2

Advance summary and potential significance to field

The authors have adequately answer my main concerns and the manuscript seems to me to be suitable for publication.

Second revision

Author response to reviewers' comments

Reviewer 1: I commend the authors for re-analyzing their data in response to the comment and for identifying a novel and intriguing role of SOS1 as a potential regulator of ERK wave rhythm. This is a valuable addition to the understanding of collective migration dynamics and suggests an interesting layer of signaling regulation.

Yet the impact of SOS1 depletion on collective migration based on the metric used, appear based on the metric used marginal albeit significant. I do understand that the major focus is on the migration of the follower,

This said, the idea that SOS1 affects the rhythm of ERK waves is intriguing, it would be helpful to comment/speculate how this alteration translates into measurable changes in migration efficiency, or whether the impact is more subtle. If the latter, a brief discussion highlighting this nuance would improve the mechanistic clarity of the manuscript and help readers appreciate the functional relevance of ERK wave rhythm in this context.

We thank the reviewer for the thoughtful comment and for highlighting the intriguing connection between ERK wave rhythm and collective migration. As noted, SOS1 depletion results in a statistically significant but relatively modest reduction in migration speed. We agree that this suggests a more subtle role for SOS1 in coordinating collective movement, likely through modulation of the ERK wave rhythm rather than through a direct impact on migration efficiency.

In the revised manuscript (Discussion, lines 302-311), we have added a paragraph to address this point. We hypothesize that the prolonged ERK wave cycle in SOS1-deficient cells arises from a higher threshold for ligand-induced RAS-ERK activation, requiring greater accumulation of growth factors to initiate signaling. This elevated threshold delays wave initiation and propagation, thereby reducing the temporal precision of ERK signaling. While this mechanism does not drastically impair migration speed, it may subtly affect the coherence and timing of cell group dynamics—factors not fully captured by our current migration metrics. We believe this nuanced effect highlights the functional relevance of ERK wave rhythm in maintaining coordinated collective behavior.

Reviewer 2: The authors have adequately answer my main concerns and the manuscript seems to me to be suitable for publication.

We thank the reviewer for the positive feedback and are pleased that our revisions have addressed the concerns. We appreciate your support for the publication of our work.

Third decision letter

MS ID#: jcs.263779R2

MS Title: Front-biased activation of Ras-Rab5-Rac1 loop coordinates collective cell migration.

Authors: Yuya Jikko; Eriko Deguchi; Kimiya Matsuda; Naoya Hino; Shinya Tsukiji; Michiyuki Matsuda; Kenta Terai

Article Type: Research Article

Dear Dr Terai,

I am happy to tell you that your manuscript has been accepted for publication in Journal of Cell Science, pending standard publication integrity checks.